# LILO: LEARNING INTERPRETABLE LIBRARIES BY COMPRESSING AND DOCUMENTING CODE

**Gabriel Grand**[1,2]    **Lionel Wong**[2]    **Maddy Bowers**[1]    **Theo X. Olausson**[1]    **Muxin Liu**[3]
**Joshua B. Tenenbaum**[1,2]    **Jacob Andreas**[1]
[1]MIT CSAIL    [2]MIT Brain and Cognitive Sciences    [3]Harvey Mudd College

## ABSTRACT

While large language models (LLMs) now excel at code *generation*, a key aspect of software development is the art of *refactoring*: consolidating code into libraries of reusable and readable programs. In this paper, we introduce LILO, a neurosymbolic framework that iteratively synthesizes, compresses, and documents code to build libraries tailored to particular problem domains. LILO combines LLM-guided program synthesis with recent algorithmic advances in automated refactoring from STITCH: a symbolic compression system that efficiently identifies optimal $\lambda$-abstractions across large code corpora. To make these abstractions *interpretable*, we introduce an auto-documentation (AutoDoc) procedure that infers natural language names and docstrings based on contextual examples of usage. In addition to improving human readability, we find that AutoDoc boosts performance by helping LILO's synthesizer to interpret and deploy learned abstractions. We evaluate LILO on three inductive program synthesis benchmarks for string editing, scene reasoning, and graphics composition. Compared to existing methods—including the state-of-the-art library learning algorithm DreamCoder—LILO solves more complex tasks and learns richer libraries that are grounded in linguistic knowledge.

## 1 INTRODUCTION

Large language models (LLMs) are growing highly adept at programming in many settings: completing partially-written code (Chen et al., 2021; Fried et al., 2022; Li et al., 2023), conversing with human programmers (Austin et al., 2021; Nijkamp et al., 2023), and even solving competition-level programming puzzles (Hendrycks et al., 2021; Li et al., 2022; Haluptzok et al., 2022; OpenAI, 2023). However, beyond solving the immediate task at hand, software engineers are concerned with building libraries that can be applied to entire problem domains. To this end, a key aspect of software development is the art of *refactoring* (Brown et al., 1998; Abrahamsson et al., 2017): identifying *abstractions* that make the codebase more concise (consolidating shared structure), reusable (generalizing to new tasks), and readable (intuitive to other programmers). Solving this multi-objective optimization will require broadening the scope of existing code completion tools—which are already used by millions of programmers—to approach the problem of *library learning*.

In this paper, we combine language models with recent algorithmic advances in automated refactoring from the programming languages (PL) literature to learn libraries of reusable function abstractions. We introduce LILO, a neurosymbolic framework for **Library Induction from Language Observations**, which consists of three interconnected modules (Fig. 1):

- A **dual-system synthesis** module, which searches for solutions to programming tasks using two distinct methods: *LLM-guided search* imbues the system with strong domain-general priors, and *enumerative search* enables the discovery of domain-specific expressions;
- A **compression** module, which identifies useful abstractions from the existing solution set via STITCH (Bowers et al., 2023), a high-performance symbolic compression system;
- An **auto-documentation (AutoDoc)** module, which generates human-readable function names and docstrings, yielding better interpretability and improving downstream LLM-guided search.

---

Correspondence to gg@mit.edu. Code for this paper is available at: github.com/gabegrand/lilo.

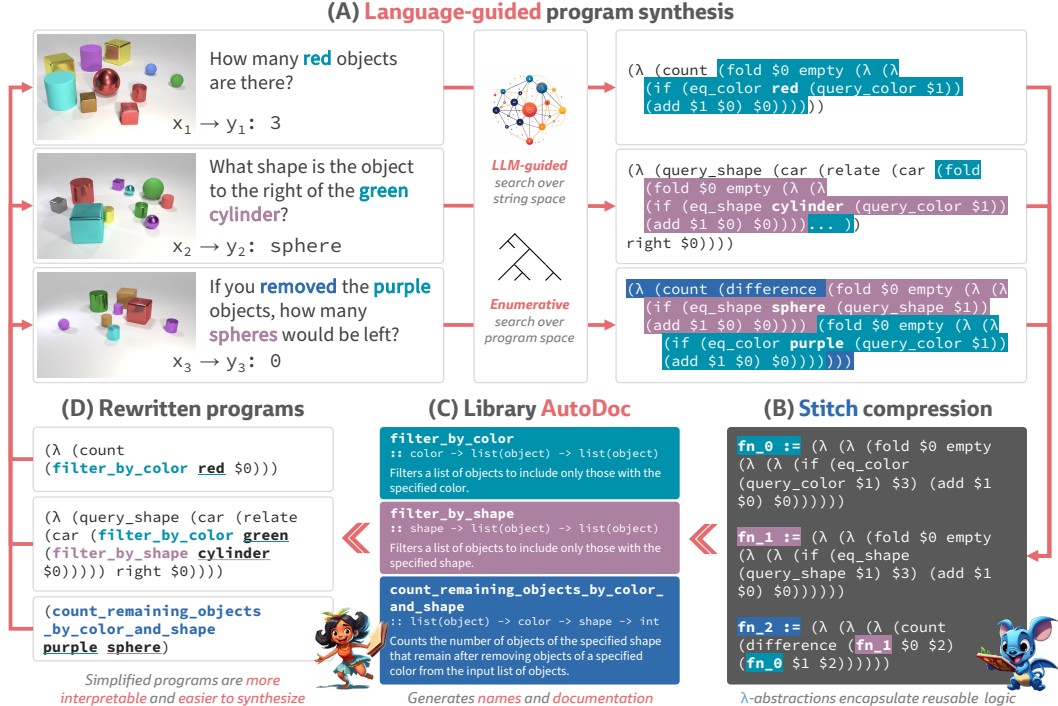

Figure 1: **Overview of the LILO learning loop.** (A) LILO synthesizes programs based on natural language task descriptions using a dual-system search model. To refactor a set of program solutions, LILO integrates a compression algorithm called STITCH (B; Bowers et al., 2023) with LLM-generated auto-documentation (C) to produce an interpretable library of λ-abstractions. This search-compress-document loop simplifies the structure of program solutions (A vs. D), making it easier to solve more complex tasks on future iterations.

Our architecture draws inspiration from DREAMCODER (Ellis et al., 2021), an iterative Wake-Sleep algorithm that alternates between searching for solutions to programming tasks (*Wake* phase) and refactoring shared abstractions into a library (*Sleep* phase) that in turn helps to guide search. Unlike standard deep learning approaches, DreamCoder can make strong generalizations from just a handful of examples, and the model's conceptual knowledge is represented symbolically in the learned library. However, DreamCoder's search procedure is extremely computationally intensive, requiring more than two CPU-*months* just to learn a single domain (see Ellis et al., 2021, Apx. J). Much of this search time is spent "getting off the ground": discovering a basic set of abstractions that human programmers typically already know, or might be able to grok quickly based on having solved problems in other domains. Moreover, DreamCoder libraries are not necessarily interpretable, requiring both domain expertise and knowledge of lambda calculus to decipher.

To address these issues, LILO leverages LLMs in two novel ways: (1) to expedite the discovery of program solutions during search, and (2) to improve the interpretability of learned libraries through documentation. We evaluate LILO against a language-guided DreamCoder variant on three challenging program synthesis domains: **string editing** with regular expressions (Andreas et al., 2018), **scene reasoning** on the CLEVR dataset (Johnson et al., 2017), and **graphics composition** in the 2D Logo turtle graphics language (Abelson & diSessa, 1986). On all three domains, LILO solves more tasks than DreamCoder and learns empirically richer libraries containing abstractions that are intractable to discover with existing methods. For instance, LILO learns the concept of a *vowel* (Fig. 2), which is a key stepping-stone in the string editing domain that would otherwise require searching over $26^5$ possible disjunctions of character primitives. Unlike LLM-only baselines—which can solve similar tasks—LILO compresses this knowledge into symbolic abstractions that are useful for both traditional search methods and LLM-guided synthesis. Key to this neurosymbolic integration is our AutoDoc module, which not only improves interpretability, but also helps the LLM synthesizer use the library more effectively. LILO illustrates how ideas and tools from the PL community can be integrated with recent breakthroughs in language models, representing a new evolution in a long line of work in inductive program synthesis, which we review below.

## 2 PRELIMINARIES: PROGRAM SEARCH AND LIBRARY LEARNING

**Program synthesis.** In inductive program synthesis (Gulwani et al., 2017), we are given a library of primitives $\mathcal{L} = \{f_1, f_2, \ldots\}$ that forms a **domain-specific language (DSL)**. For a given programming task $t = \{(x_i, y_i)\}$ specified as a set of input-output pairs, the goal is to find a program $\pi : \forall_i \pi(x_i) = y_i$ that correctly maps all inputs to outputs, denoted $\pi \vdash t$. However, a typical task admits many such solutions that will not necessarily generalize (for instance, a simple lookup table). To address this inherent under-specification, concern is given to finding an *optimal* program $\hat{\pi} \vdash t$ with respect to descriptive complexity (Solomonoff, 1964; Kolmogorov, 1965; Chaitin, 1966). This optimization is naturally framed in terms of probabilistic inference:

$$\arg\max_{\pi} \log p(\pi \mid t, \mathcal{L}) = \arg\max_{\pi} \left[ \log p(t \mid \pi) + \log p(\pi \mid \mathcal{L}) \right] \tag{1}$$

In a typical setting, the likelihood $p(t \mid \pi) \triangleq \mathbb{1}_{\pi \vdash t}$ is computed via program execution, while the prior $p(\pi \mid \mathcal{L}) \triangleq \prod_{f \in \pi} p(f \mid \mathcal{L})$ is defined under a probabilistic context free grammar (PFCG; Johnson, 1998) that assigns a weight $0 \leq \theta \leq 1$ to each production rule. This is equivalent to a weighted *description length* prior, where longer programs have lower probability.

This formulation highlights the central challenge of program synthesis: historically, approaches to Eq. 1 have inevitably involved enumerative search through a combinatoral space of programs. A range of techniques have been proposed to improve search tractability, including type-directed synthesis (Polikarpova et al., 2016), Monte Carlo approximation (Liang et al., 2010; Allamanis et al., 2018; Shin et al., 2019), and neural network guidance (Gulwani et al., 2015; Balog et al., 2017; Parisotto et al., 2017; Devlin et al., 2017; Nye et al., 2019; Ellis et al., 2021). However, even with these methods, traditional program synthesis hinges critically on DSL design. Omission of key primitives can make complex tasks unsolvable, while inclusion of extraneous primitives can make search intractable. Consequently, DSL engineering is a painstaking process that requires significant expertise to anticipate common patterns across tasks in a domain.

**Library learning.** While classical approaches focus on synthesizing the best program for a task specification given a fixed DSL (as in Eq. 1), programmers in the wild are typically concerned with solving entire problem domains. Given the difficulty of manual DSL engineering, a natural evolution is to include $\mathcal{L}$ itself as part of the optimization problem. This is the main intuition behind *library learning* methods (Liang et al., 2010; Dechter et al., 2013; Lake et al., 2015; Shin et al., 2019; Lázaro-Gredilla et al., 2019; Ellis et al., 2018; 2021), which start with a collection of tasks $\mathcal{T} = \{t_1, t_2, \ldots\}$ and a base library $\mathcal{L}_0$, and jointly infer an expanded library $\mathcal{L} = \mathcal{L}_0 \cup \{f_1^*, \ldots, f_k^*\}$ that includes additional **abstractions** $f^*$ built from $\mathcal{L}_0$ (Fig. 1B) and programs $\Pi = \{\pi_1, \pi_2, \ldots\}$ written in terms of $\mathcal{L}$:

$$\arg\max_{\Pi, \mathcal{L}} \log p(\Pi, \mathcal{L} \mid \mathcal{T}, \mathcal{L}_0) = \arg\max_{\Pi, \mathcal{L}} \left[ \sum_{t \in \mathcal{T}} \log p(t \mid \pi_t) + \log p(\pi_t \mid \mathcal{L}) \right] + \log p(\mathcal{L} \mid \mathcal{L}_0) \tag{2}$$

This objective carries over the program prior and likelihood from Eq. 1, but introduces a distribution over libraries $p(\mathcal{L} \mid \mathcal{L}_0)$, typically also defined in terms of description length. Intuitively, Eq. 2 is optimized by inventing abstractions that are both *reusable*, simplifying the solutions to multiple tasks in $\mathcal{T}$; and *concise*, ideally building on one another hierarchically so as to share logic. Ellis et al. (2021) approximate Eq. 2 via coordinate ascent, alternating between a **search step**, which holds the library fixed and searches for task solutions $\Pi$, and a **refactoring step**, which extracts common structure from the solution set to update $\mathcal{L}$. The tractability of this approach hinges critically on the ability to do efficient refactoring, which we discuss further in §3.

**Leveraging language guidance.** Given the size of the search space, generic priors such as description length are not always sufficient to solve Eq. 1; for this reason, a line of work considers natural language task descriptions $d_t$ as an additional source of learning signal (Manshadi et al., 2013; Raza et al., 2015; Shin et al., 2019). Traditionally, making use of such descriptions has required learning a domain-specific semantic parsing model (Liang et al., 2011; Artzi & Zettlemoyer, 2013; Chen & Mooney, 2011). More recent work (Rahmani et al., 2021; Yin et al., 2022; Zelikman et al., 2022) uses LLMs, which excel when $\mathcal{L}$ resembles a common programming language that is well-represented in pretraining.

In library learning settings—where $\mathcal{L}$ is novel by construction—it is currently less clear how to leverage language. In LAPS (Language for Abstraction and Program Search), Wong et al. (2021)

generalize Eq. 2 to condition on $d_t$ by fitting an inverted "program-to-language" translation model. However, learning this mapping from scratch necessitates the use of a small alignment model (*IBM Model 4*; Brown et al., 1993) that makes strict token-to-token decomposition assumptions. In LILO, we take the opposite approach: we start with a large model that already has strong priors over the joint distribution of language and code; then, we adapt the *library* to resemble this distribution by building up contextual examples and documentation. In contrast to simply picking a more common $\mathcal{L}$ (e.g., Python) to work in, this procedure enables us to *learn* a new $\mathcal{L}$ on-the-fly that is both optimized for the domain and grounded in natural language.

## 3 LILO: LIBRARY INDUCTION WITH LANGUAGE OBSERVATIONS

Our method, LILO, aims to combine the strong inductive search biases encoded by LLMs with the key ideas from classical methods in library learning—namely, the ability to discover new symbolic abstractions through refactoring. Algorithmically, LILO (Alg. 1) has a similar structure to existing approaches that optimize Eq. 2 via coordinate ascent, alternating between search and refactoring. However, unlike prior work, our use of LLMs for search necessitates an additional step—AutoDoc— to render the learned abstractions legible to the synthesizer. We detail each of these modules below.

**Dual-system program search (Fig. 1A).** Inspired by dual process theories of cognition (Sloman, 1996; Evans, 2003; Kahneman, 2011), LILO is equipped with two distinct search procedures. We use a LLM as a "fast" approximate search model in string space that leverages strong inductive biases learned in pretraining. After this first pass, we perform "slow" enumerative search in program space, using a task-conditioned PCFG inferred by a "recognition network" for guidance. We focus here on LLM-guided synthesis and refer to Ellis et al., 2021 (Apx. E and I) for details on enumeration.

Formally, we write $p_{\text{LLM}}(y \mid x)$ to denote the distribution over strings $y$ produced by a language model prompted with string $x$. Then, for some target task $\hat{t}$, our goal is to approximate the conditional distribution over programs

$$p(\pi_{\hat{t}} \mid \mathcal{L}, \Pi, d_{\hat{t}}) \approx p_{\text{LLM}}(\langle \pi_{\hat{t}} \rangle \mid \underbrace{\langle f \mid f \in \mathcal{L} \rangle}_{\text{library functions}} \circ \underbrace{\langle (d_t, \pi_t) \mid \pi_t \sim \Pi \rangle}_{\text{program examples}} \circ \underbrace{\langle d_{\hat{t}} \rangle}_{\text{task desc.}}) \tag{3}$$

where $\langle \ldots \rangle$ and $\circ$ denote string serialization and concatenation, respectively. To sample from the distribution in Eq. 3, we procedurally construct few-shot prompts consisting of three parts: (1) A library description that enumerates the available primitives and any learned abstractions, (2) a set of exemplars consisting of description-solution pairs $(d_t, \pi_t) \sim \Pi$ sampled from the set of solved tasks, and (3) a description of the target task $d_{\hat{t}}$. For each completion, we run parsing, type inference, and execution checks to identify valid programs that solve the target task. (Appendix A.1 illustrates the composition of a typical prompt; A.3 contains additional details on how examples are sampled.)

**Refactoring via Stitch compression (Fig. 1B).** As the learner solves more tasks, the solution set will grow to contain many recurring program fragments that we wish to refactor into a set of reusable abstractions. In library learning systems that rely on enumeration, refactoring improves search efficiency by avoiding the need to rediscover key building blocks for each new task. Analogously, in

---

**Algorithm 1** Library learning loop with LILO

1: **function** LILOLEARNING($\mathcal{L}_0, \mathcal{T}$)
2:     $\mathcal{L} \leftarrow \mathcal{L}_0$         ▷ *Initialize library with base DSL*
3:     $\Pi \leftarrow \{t : \emptyset \mid t \in \mathcal{T}\}$         ▷ *Initialize task solution set*
4:     **for** $i = 1, \ldots, N$ **do**
5:         **for** $t \in \mathcal{T}$ **do**         ▷ *Run LLM Solver*
6:             $\Pi_t \leftarrow \Pi_t \cup \text{LLM}(\text{TaskPrompt}(\mathcal{L}, \Pi, d_t))$
7:         $\Pi \leftarrow \Pi \cup \text{SEARCH}(\mathcal{L}_i, \mathcal{T})$     ▷ *Run enumerative search (skipped in ✂ Search)*
8:         $\{f_1^*, \ldots, f_k^*\} \leftarrow \text{COMPRESS}(\mathcal{L}, \Pi, k)$     ▷ *Generate new abstractions*
9:         $\mathcal{L} \leftarrow \mathcal{L}_0 \cup \{f_1^*, \ldots, f_k^*\}$
10:         $\Pi \leftarrow \text{REWRITE}(\mathcal{L}, \Pi)$
11:         **for** $\alpha \in \{f_1^*, \ldots, f_k^*\}$ **do**     ▷ *Document abstractions (skipped in ✂ AutoDoc)*
12:             $\mathcal{D} \leftarrow \text{LLM}(\text{AutoDocPrompt}(\mathcal{L}, \Pi, \alpha))$
13:             $\mathcal{L} \leftarrow \text{add\_docs}(\mathcal{L}, \alpha, \mathcal{D})$
14:     **return** $\mathcal{L}, \Pi$     ▷ *Return final library and task solutions*

---

*(A) Anonymous abstractions from **Stitch***

```
fn_42 :: tsubstr
(or 'a' (or 'e' (or 'i' (or 'o' 'u'))))
fn_43 :: tfullstr -> tsubstr -> tsubstr -> tfullstr
(λ (λ (λ (flatten (map (λ (if (match $1 $0) $2 $0)) (split empty_string $2))))))
    ⋮
fn_51 :: tsubstr -> tsubstr -> tfullstr -> tfullstr
(λ (λ (λ (fn_46 $0 (fn_44 $0 $1 $2) (not fn_42)))))
```

*Please write a human-readable name and description for* **fn_42**. *Here are some examples of its usage:*

*-- if there is vowel replace that with s*
(λ (**fn_43** $0 's' **fn_42**))

*-- if there is consonant add s after that*
(λ (**fn_49** 's' (not **fn_42**) $0))

*-- if the word starts with vowel replace that with u c*
(λ (**fn_46** $0 (**fn_44** $0 'c' 'u') **fn_42**))

*Please write a human-readable name and description for* **fn_43**. *Here are some examples of its usage:*

*-- if there is d replace that with y*
(λ (**fn_43** $0 'y' 'd'))

*-- if there is i replace that with k t*
(λ (**fn_43** $0 (concat 'k' 't') 'i'))

*-- if there is consonant replace that with p*
(λ (**fn_43** $0 'p' (not vowel_regex)))

*Please write a human-readable name and description…*

(fn_42) **vowel_regex** :: tsubstr
Regular expression that matches any vowel ('a', 'e', 'i', 'o', 'u'). Used in various functions to identify and modify words based on vowel presence and position.
(or 'a' (or 'e' (or 'i' (or 'o' 'u'))))

(fn_43) **replace_substr** :: tfullstr -> tsubstr -> tsubstr -> tfullstr
Replaces all instances of a given substring $1 in a full string $0 with another substring $2...
(λ (λ (λ (flatten (map (λ (if (match $1 $0) $2 $0)) (split empty_string $2))))))

(fn_44) **replace_first_occurrence**

(fn_47) **replace_if_match_substring**

(fn_51) **replace_consonant_with_substring** :: tsubstr -> tsubstr -> tfullstr -> tfullstr
Replaces the first occurrence of a consonant at the beginning of a given full string with a specified substring...
(λ (λ (λ (replace_if_match_substring $0 (replace_first_occurrence $0 $1 $2) (not vowel_regex)))))

*(B) **LILO** AutoDoc prompt sequence*      *(C) Human-readable library*

Figure 2: **LILO library auto-documentation (AutoDoc) workflow in the REGEX domain.** For each Stitch abstraction (A), we prompt an instruction-tuned LLM with usage examples from solved tasks (B) to generate a human-readable name and description (C). The chat-style structure of AutoDoc allows naming choices to cascade sequentially; e.g., replace_consonant_with_substring (fn_51) refers back to vowel_regex (fn_42) and other named abstractions in a consistent and interpretable manner.

LILO, refactoring makes the generation task easier: a LLM equipped with a library of abstractions can deploy entire blocks of code with just a few tokens. Additionally, refactoring helps to mitigate LLM context window limits by minimizing the description length of the few-shot examples.

Various algorithms for refactoring have been proposed using combinatory logic (Liang et al., 2010), tree substitution grammars (Allamanis & Sutton, 2014; Allamanis et al., 2018; Shin et al., 2019), version spaces (Lau & Weld, 1998; Ellis et al., 2021), and e-graphs (Cao et al., 2023). In LILO, we cast refactoring as a *compression problem* over a corpus of programs

$$f^* = \underset{\mathcal{L}}{\text{COMPRESS}}(\Pi) = \arg\min_f |f| + \sum_{\pi \in \Pi} |\underset{\mathcal{L} \cup \{f\}}{\text{REWRITE}}(\pi)| \tag{4}$$

where the goal is to identify abstractions with minimal description length $|f|$ that facilitate efficient rewriting of $\Pi$. However, performing even a single round of compression as in Eq. 4 necessitates an efficient search strategy. In LILO, we leverage recent algorithmic advances from STITCH (Bowers et al., 2023): a symbolic compression system that uses branch-and-bound search to identify reusable abstractions in large datasets of lambda calculus programs. While Bowers et al. demonstrate that Stitch is 1000–10000x faster and 100x more memory efficient than DreamCoder's compression algorithm, prior analyses were limited to static corpora of ground truth programs. In LILO, we deploy Stitch for the first time in a synthesis loop and find it similarly performant, typically running in seconds on a single CPU. These efficiency improvements enable us to re-derive the entire library from $\mathcal{L}_0$ at every iteration (Alg. 1 line 9). While many abstractions remain stable across iterations, this "deep refactoring" allows LILO to discard suboptimal abstractions discovered early in learning.

**Library auto-documentation (Fig. 1C).** Unlike traditional program synthesis methods, language models draw inferences about the semantics of programs from their lexical content. In other words, LLMs (like human programmers) care what functions are named. However, PL tools are typically not equipped to write human-readable function names, instead outputting anonymous lambda abstractions (in Stitch, these are represented numerically; e.g., fn_42). In early experiments, we observed that naively providing a LLM with Stitch abstractions measurably degraded its ability to solve tasks (§4.1). We hypothesized that rewriting program examples in terms of these anonymous

lambdas during the compression step (Eq. 4) was acting as a form of *code obfuscation* (Srikant et al., 2021; Zeng et al., 2022; Miceli-Barone et al., 2023).

Motivated by these findings, as part of LILO, we introduce a *library auto-documentation* (AutoDoc) procedure that writes human-readable names and docstrings for abstractions generated by Stitch. AutoDoc leverages the fact that LLMs are naturally suited to code deobfuscation (Lachaux et al., 2021; Sharma et al., 2022; Cambronero et al., 2023). During AutoDoc, we sequentially prompt an instruction-tuned LLM to produce a human-readable name and docstring for each abstraction in the library. Fig. 2 gives an overview of this workflow (the full prompt is reproduced in A.2). In this example in the REGEX domain, the LLM has solved some problems that require vowel substitutions. During compression, Stitch pulls out the expression `(or 'a' (or 'e' (or 'i' (or 'o' 'u'))))` for occurring commonly in the solution set and defines it as an anonymous arity-0 function (i.e., a constant). Subsequently, AutoDoc names this abstraction `vowel_regex`, which forms the basis for more complex expressions. For instance, *consonant* is expressed as `(not vowel_regex)`, which in turn supports an abstraction for consonant replacement. In §4, we explore how AutoDoc benefits downstream synthesis performance, yielding both richer and more interpretable libraries.

## 4 EXPERIMENTS AND RESULTS

**Domains.** Our goal is to build a system that can develop expertise in novel technical domains and leverage natural language guidance to bootstrap learning. We evaluate our approach on three inductive program synthesis domains: string editing (REGEX), scene reasoning (CLEVR), and graphics composition (LOGO). Detailed descriptions and metrics pertaining to each domain are provided in B.1. These three domains were introduced in LAPS (Wong et al., 2021) as more challenging versions of the domains evaluated in DreamCoder and our evaluations are directly comparable to prior results (B.3). Following Wong et al., we use synthetic ground truth task descriptions in our primary experiments and evaluate on a noisier set of human language annotations in B.4.

**Experiment setup.** Our experiments are designed to simulate a "lifelong learning" setting where the learner starts from a small seed set of simple examples that it must generalize to solve a broader set of training tasks that range in complexity. Performance is measured as the percentage of tasks solved from an i.i.d. test set. We sequentially perform two experiments that test different aspects of models' learning. First, in **online synthesis**, each model runs for a fixed number of iterations, continually updating its library (if applicable) and attempting to solve test tasks. These experiments serve as a general benchmark of language-guided synthesis capabilities. Next, in **offline synthesis**, we freeze the final library $\mathcal{L}_f$ from each online synthesis run and perform enumerative search with no language guidance for a fixed time budget. (To compare against non-library learning baselines, we run STITCH to generate $\mathcal{L}_f$ *post-hoc*.) We hold the hyperparameters of the search fixed so that performance depends entirely on $\mathcal{L}_f$ and not on the original model. Thus, the offline synthesis evaluations provide a controlled comparison of the off-the-shelf utility of different learned libraries in the absence of language guidance. Throughout, comparisons between models are expressed in terms of absolute percentage point changes in mean solve rates on the test set.

**Implementation details.** For LLM-guided search, we queried OpenAI's Codex model (`code-davinci-002`)[1] with up to 4 prompts per task, sampling 4 completions per prompt. For AutoDoc (which requires significantly fewer queries) we found that OpenAI's newer instruction-tuned models (`gpt-3.5-turbo` and `gpt-4`) better adhered to the AutoDoc task and schema. Further implementation details can be found in Appendices A.4–A.5.

### 4.1 EVALUATION OF SYNTHESIS PERFORMANCE

We compare LILO against two baseline models: a language-guided DreamCoder variant from Wong et al. (2021) and a language model (LLM Solver) that does not perform library learning (Fig. 3). To study the effects of the different LILO components, we introduce ablated variants that remove the enumerative search and/or AutoDoc steps. To ensure fair comparison and improve overall runtimes, Stitch is used as the compression module for all library learning models, including the DreamCoder baseline. Tab. 1 gives the full breakdown of task solution rates.

---

[1]We accessed Codex through OpenAI's free beta program for researchers, which saved thousands of USD over the project lifetime (see C.1) and afforded higher rate limits than paid GPT models. To preserve reproducibility, we make all LLM generations available at: `github.com/gabegrand/lilo`.

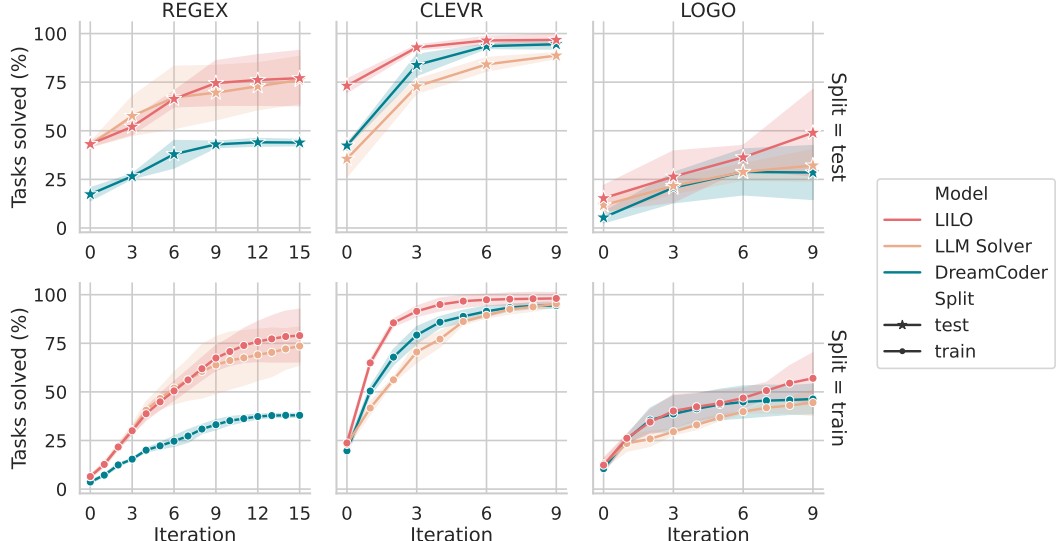

Figure 3: **Learning curves during online synthesis.** Within each plot, the x-axis tracks the experiment iteration and the y-axis shows the percent of tasks solved (top = test, bottom = train). Error bars show standard deviation across 3 randomly-seeded runs.

| | REGEX | | | CLEVR | | | LOGO | | |
|---|---|---|---|---|---|---|---|---|---|
| MODEL | *max* | *mean* | *std* | *max* | *mean* | *std* | *max* | *mean* | *std* |
| DreamCoder | 45.60 | 43.93 | 1.53 | 97.09 | 94.50 | 2.44 | 36.94 | 28.53 | 13.79 |
| LLM Solver | 90.00 | 76.13 | 12.04 | 90.29 | 88.67 | 1.48 | 41.44 | 32.13 | 8.07 |
| LLM Solver (+ Search) | 91.20 | 76.60 | 13.02 | 97.09 | 96.44 | 0.56 | 45.05 | 37.84 | 6.80 |
| LILO (✂ Search / AutoDoc) | 59.40 | 53.20 | 5.38 | 93.20 | 85.76 | 9.72 | 45.05 | 21.02 | 20.88 |
| LILO (✂ Search) | 63.80 | 62.93 | 1.50 | 94.17 | 88.03 | 8.26 | 30.63 | 21.02 | 9.46 |
| LILO | **93.20** | **77.07** | 14.14 | **99.03** | **96.76** | 3.12 | **73.87** | **48.95** | 22.15 |
| Base DSL | 22.00 | 22.00 | 0.00 | 29.13 | 29.13 | 0.00 | 0.90 | 0.90 | 0.00 |
| DreamCoder | 42.00 | 41.60 | 0.40 | 94.17 | 91.59 | 2.97 | 36.04 | 30.63 | 7.85 |
| LLM Solver[*] | 48.60 | 43.00 | 5.17 | 91.26 | 89.64 | 2.02 | 36.04 | 27.33 | 7.56 |
| LLM Solver (+ Search)[*] | 63.40 | 55.67 | 7.51 | 91.26 | 89.00 | 3.92 | 28.83 | 27.63 | 1.04 |
| LILO (✂ Search / AutoDoc) | 60.80 | 50.73 | 8.85 | 95.15 | 93.85 | 2.24 | **51.35** | 30.63 | 18.22 |
| LILO (✂ Search) | 57.60 | 56.20 | 2.25 | **96.12** | **95.79** | 0.56 | 28.83 | 26.13 | 3.25 |
| LILO | **71.40** | **64.27** | 6.31 | **96.12** | 92.56 | 6.17 | 50.45 | **41.14** | 8.66 |

Table 1: **Task solution rates for online (upper) and offline (lower) synthesis experiments.** We report the best (*max*), average (*mean*), and standard deviation (*std*) test solve rates across model runs. In each mean column, results within $1\sigma$ of the best (**bold**) result are underlined. [*]Asterisk indicates $\mathcal{L}_f$ computed *post-hoc*.

**LLMs facilitate effective search over lambda calculus programs.** Our first question is whether the LLM-based search procedure introduced in §3 can match the accuracy of enumerative search. Compared to DreamCoder, we find that the LLM Solver, with no library learning, performs significantly better on REGEX ($+32.20$), slightly worse on CLEVR ($-5.83$), and comparably on LOGO ($+3.60$). The improvements on REGEX are primarily attributable to LLMs' ability to generate expressions for concepts like vowel and consonant that invoke prior knowledge.

**LILO achieves the strongest overall performance.** As observed in Fig. 3, LILO significantly outperforms DreamCoder on REGEX ($+33.14$) and LOGO ($+20.42$). It also achieves small improvements on CLEVR ($+2.26$), though the DreamCoder baseline is already quite high for this domain ($\mu = 94.50$, $\sigma = 2.44$). Moreover, LILO also improves on the LLM Solver baseline by $+0.94$ (REGEX), $+8.09$ (CLEVR), and $+16.82$ (LOGO). A key advantage of LILO is the ability to perform enumerative search, which aids in the discovery novel programs that differ structurally from existing solutions in the LLM prompts. To isolate the effects of search, we ran an ablation [LILO (✂ Search)] as well as an augmented baseline [LLM Solver (+ Search)]. We find that enumerative search is most helpful on LOGO, which requires certain domain-specific program structures (e.g., how to draw a "snowflake" or "staircase"; see Fig. 5) that are difficult to infer from language alone.

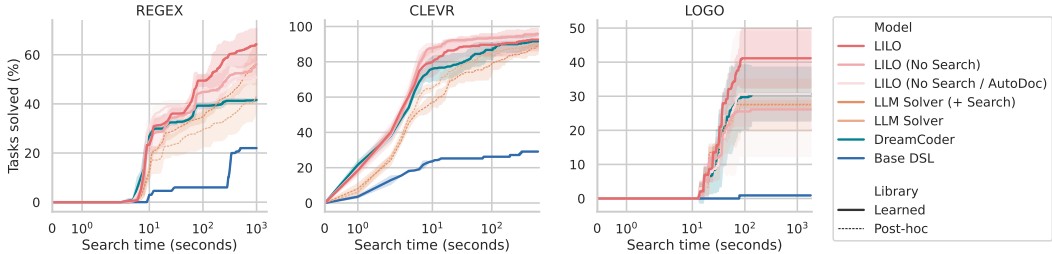

Figure 4: **Evaluating library quality via offline synthesis.** We run a timed enumerative search (x-axis; note the log-scale) with the final library $\mathcal{L}_f$ learned by each model in online synthesis or inferred *post-hoc*. In this setting, LILO's $\mathcal{L}_f$ expedites discovery of test task solutions (y-axis) even without language guidance.

**Auto-documentation unlocks effective contextual usage of abstractions.** Early experiments interfacing the LLM Solver with Stitch [Tab. 1, LILO (✂ Search / AutoDoc)] revealed a puzzling finding: providing the LLM with abstractions did not help—and in some cases, significantly hurt—downstream synthesis performance. Relative to the LLM Solver baseline, we observed solution rate changes of $-30.60$ (REGEX), $-2.91$ (CLEVR), and $-11.11$ (LOGO) after introducing Stitch compression. Qualitative inspection found that GPT struggled to deploy anonymous abstractions in contextually-appropriate ways, which motivated our exploration of de-obfuscation methods (§3). After introducing AutoDoc [Tab. 1, LILO (✂ Search)], we see mean improvements of $+9.73$ (REGEX) and $+2.27$ (CLEVR) over the naive condition. On LOGO, AutoDoc does not improve performance ($+0.00$). We attribute this to the occurrence of semantic errors, which we analyze further in §4.2.

**LILO rivals symbolic search in terms of computational efficiency.** Compared to enumerative search, we find that we are able to achieve faster overall wall clock runtimes with LLM-based search due to orders of magnitude better sample efficiency (see C.1). Indeed, in terms of dollar cost, we estimate that one round of LLM synthesis is equivalent to 48 CPU-hours of DreamCoder search. Of course, LLM-based and enumerative search are not mutually exclusive: our main LILO variant integrates both of these procedures and achieves the highest solve rates of all models we evaluated. Our findings make a strong case that LLMs can reduce the need for exhaustive search in synthesis domains where language annotations are available.

**LILO libraries generalize well even in the absence of language.** In our offline synthesis experiments, we tested each model's final library $\mathcal{L}_f$ in an off-the-shelf setting with no language guidance. Fig. 4 shows the results of these evaluations (metrics are given in Tab. 1, lower). As the baseline for each domain, we measure synthesis performance in $\mathcal{L}_0$ (Base DSL). As expected, we can significantly outperform $\mathcal{L}_0$ using library learning: DreamCoder's $\mathcal{L}_f$ improves on the $\mathcal{L}_0$ solve rates by $+19.6$ (REGEX), $+62.46$ (CLEVR), and $+29.73$ (LOGO). Moreover, in each domain, LILO's $\mathcal{L}_f$ improves further on DreamCoder, showing solution rate gains of $+42.27$ (REGEX), $+63.43$ (CLEVR), and $+43.24$ (LOGO) over $\mathcal{L}_0$. LILO's $\mathcal{L}_f$ also outperforms libraries derived *post-hoc* from the two LLM Solver baselines, highlighting the benefits of performing compression and documentation in-the-loop. As these results demonstrate, LILO learns high-quality libraries that generalize well to downstream synthesis tasks and can be used even in the absence of language guidance.

## 4.2 QUALITATIVE ANALYSIS OF LIBRARY ABSTRACTIONS

In all three domains, the libraries learned by LILO exhibit examples of compositional and hierarchical reuse. For instance, in the LOGO library (Fig. 5 and B.2.3), the top abstraction is a general method for drawing polygons that is invoked by several higher-level abstractions. Similarly, the CLEVR library (Fig. 1 and B.2.2) contains two layers of abstractions: a lower layer implements `filter` operations over size, color, shape, and material attributes that support a higher layer of more specialized abstractions for counting and filtering. These examples showcase how LILO builds on one of the main strengths of DreamCoder—the ability to bootstrap hierarchies of learned concepts—while improving the richness and interpretability of the learned libraries through auto-documentation.

While the GPT models do a remarkable job at inferring program semantics, we observe various cases where they produce unclear or even incorrect documentation. For instance, in LOGO (Fig. 5),

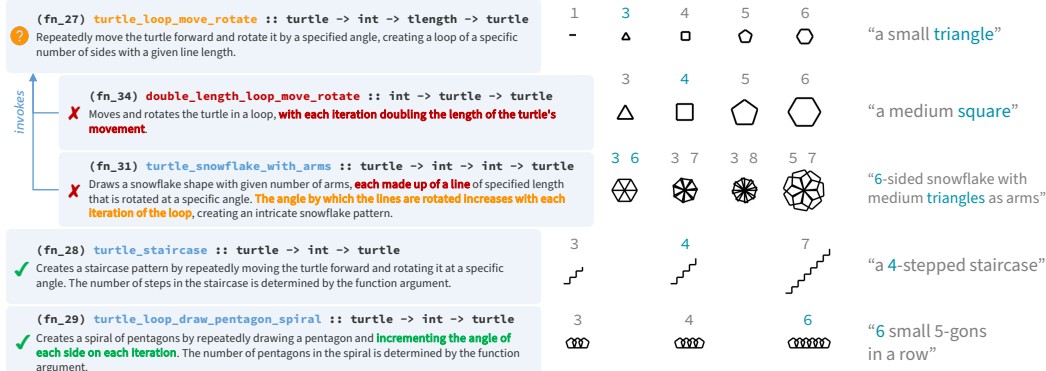

Figure 5: **Qualitative inspection of LOGO library.** Selected examples of graphics abstractions learned by LILO. Highlights indicate ambiguities (orange) and errors (red) in naming and documentation that may affect code comprehension, which we discuss below.

the two polygon functions (fn_27 and fn_34) are assigned relatively uninformative names that emphasize their implementation (looping move and rotate) but not their behavior (drawing polygons). Moreover, because AutoDoc works sequentially, it occasionally "doubles down" on particular statements that may be correct in one context but not another. For example, AutoDoc correctly notes that fn_27 works by "incrementing the angle of each side on each iteration," but this idea is ambiguous in fn_31 (*which angle?*) and incorrect in fn_34 (*the length is constant, not doubling*). In addition to affecting interpretability, these semantic errors may also impact downstream synthesis performance in LLM-guided search. Future work could adopt self-consistency and verification techniques (Wang et al., 2022; Dhuliawala et al., 2023) to improve the quality of AutoDoc generations.

## 5 DISCUSSION AND CONCLUSION

In this work, we introduced LILO, a neurosymbolic framework for learning interpretable libraries for program synthesis. In our experiments, we found that LILO performs favorably compared to both DreamCoder and an LLM-only baseline. Much of LILO's advantage comes from synergy between neural and symbolic components: LLM-guided search enables LILO to learn concepts that are intractable to discover with enumerative search; compression allows LILO to consolidate these solutions into reusable abstractions available to symbolic search; and finally, AutoDoc makes these abstractions legible for both humans and LLMs.

While LILO improves on prior library learning approaches, notably, the LLM-only baseline also demonstrates the ability to bootstrap its performance over time. This result aligns with recent successes in automated prompting (Zhou et al., 2023; Yao et al., 2022), suggesting that transformer attention can be viewed as implementing a form of *non-compressive* library learning where task-relevant information is accumulated in the prompt. However, it is unclear whether this approach will scale to large software libraries: as context length grows, key information may be "lost in the middle" or ignored due to ordering effects (O'Connor & Andreas, 2021; Lu et al., 2022; Liu et al., 2023). Accordingly, an important line of research looks to develop new strategies for equipping LLMs with long-term memory through retrieval (Wu et al., 2022; Park et al., 2023), self-reflection (Shinn et al., 2023), or combinations of both that enable learning libraries of programmatic skills in embodied environments (Wang et al., 2023). Currently, these approaches face the common challenge of determining what information to preserve, leading to a large space of *ad hoc* heuristics.

LILO offers a principled approach to the consolidation of knowledge in a lifelong learning setting, adding program compression to a growing toolkit of LLM integrations with symbolic computation (Schick et al., 2023; Wolfram, 2023). Moreover, given the generality of Stitch's algorithmic approach, extending LILO to modern imperative languages (e.g., Python) reduces to a PL research problem that is both presently tractable and technically compelling. In contrast to the view that LLMs will subsume formal accounts of programming languages, LILO offers a blueprint for collaboration between the ML and PL communities towards the longstanding goal of learning interpretable software libraries that enable solutions to novel problem domains.

ACKNOWLEDGMENTS

This work benefited greatly from discussions with colleagues in academia and industry, including Sam Acquaviva, Ekin Akyürek, Jacob Austin, Armando Solar-Lezama, and Belinda Li. We are thankful to Jack Feser for his OCaml wizardry. Finally, LILO owes a significant debt to ideas and infrastructure pioneered by Kevin Ellis, and we are deeply grateful for his feedback and guidance.

The authors gratefully acknowledge support from the MIT Quest for Intelligence, the MIT-IBM Watson AI Lab, the Intel Corporation, AFOSR, DARPA, and ONR. GG and MB are supported by the National Science Foundation (NSF) under Grant No. 2141064. GG was additionally supported by the MIT Presidential Fellowship. TXO is supported by the Herbert E. Grier (1933) and Dorothy J. Grier Fund Fellowship and the DARPA ASKEM program (award #HR00112220042). GG, MB, TXO, and JA are supported by the National Science Foundation (NSF) and Intel through NSF Grant CCF:2217064. JA is supported by NSF Grant IIS-2238240. LW and JBT received support from AFOSR Grant #FA9550-19-1-0269, the MIT-IBM Watson AI Lab, ONR Science of AI and the DARPA Machine Common Sense program. Any opinions, findings, and conclusions or recommendations expressed in this material are those of the author(s) and do not necessarily reflect the views of sponsors.

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

# Appendix

## Table of Contents

# A  METHODS

## A.1  LLM SOLVER PROMPT

Figure 6: **Anatomy of a LLM Solver prompt.** (A) Each prompt begins with a short domain description followed by an autogenerated list of the DSL primitives and their type signatures. (B) We randomly sample task solutions and their language descriptions to construct the prompt body. (C) The final line of the prompt contains a target task description for an unsolved task. (D) We sample and parse $N = 4$ completions from the LLM, filter out invalid programs, and check for task solutions.

## A.2 Auto-Documentation Prompt

For reproducibility, we provide an example of the full text of an AutoDoc prompt sequence for the REGEX domain below. The prompt is composed of multiple pieces that are sent in serial as messages to the ChatGPT interface. The sequence begins with a header message describing the DSL. For pedagogical clarity, we consider the case where every abstraction except the final one have already assigned names. Thus, the header contains a mostly-documented library with the final fn_51 remaining anonymous.

```
You are writing software documentation. Your goal is to write human-readable names for
the following library functions:

vowel_or :: tsubstr
(regex_or 'a' (regex_or 'e' (regex_or 'i' (regex_or 'o' 'u'))))
{- Matches any single vowel character ('a', 'e', 'i', 'o', 'u') using 'regex_or'
function. -}

replace_and_flatten :: tfullstr -> tsubstr -> tsubstr -> tfullstr
(lambda (lambda (lambda (regex_flatten (regex_map (lambda (regex_if (regex_match $2 $0)
$1 $0)) (regex_split $1 $2))))))
{- Replaces all instances of a given substring with another substring, and returns the
resulting string flattened into one string. The first argument is the input string, the
second argument is the substring to be replaced, and the third argument is the
substring to use instead of the replaced substring. -}

                ... <fn_44 - fn_50 omitted for concision> ...

fn_51 :: tfullstr -> tsubstr -> tsubstr -> tfullstr
(lambda (lambda (lambda (regex_flatten (regex_cons $0 (regex_cons $1 (regex_cdr
(split_string_into_list $2)))))))))
```

We then send a message prompting the LLM to document fn_51. At the end of the message, we request that the LLM encode the reply into a particular JSON format to facilitate downstream parsing.

```
Consider the following anonymous function:

fn_51 :: tfullstr -> tsubstr -> tsubstr -> tfullstr
(lambda (lambda (lambda (regex_flatten (regex_cons $0 (regex_cons $1 (regex_cdr
(split_string_into_list $2)))))))))

Here are some examples of its usage:

-- if the word starts with consonant any letter replace that with v d
(lambda (regex_if (regex_match (regex_not vowel_or) (regex_car (split_string_into_list
$0))) (fn_51 (regex_flatten (regex_cdr (split_string_into_list $0))) 'd' 'v') $0))

-- if the word starts with any letter vowel add q before that
(lambda (regex_if (regex_match vowel_or (regex_car (regex_cdr (split_string_into_list
$0)))) (fn_51 $0 (regex_car (split_string_into_list $0)) 'q') $0))

-- if the word starts with vowel replace that with u c
(lambda (regex_if (regex_match vowel_or (regex_car (split_string_into_list $0))) (fn_51
(regex_flatten (split_string_into_list $0)) 'c' 'u') $0))

            ... <additional usage examples omitted for concision> ...

Please write a human-readable name and description for `fn_51` in the JSON format shown
below.
Your `readable_name` should be underscore-separated and should not contain any spaces.
It should also be unique (not existing in the function library above).
If you cannot come up with a good name, please set `readable_name` to `null`.
```

```
{
    "anonymous_name": "fn_51",
    "readable_name": TODO,
    "description": TODO
}
```

We encountered difficulties in coaxing Codex to perform the AutoDoc task: the resulting function names were variable in quality, did not reliably capture the function semantics, and were embedded in generations that did not always adhere to the desired output specification. Instead, we take advantage of OpenAI's instruction-tuned `gpt-3.5-turbo` and `gpt-4` models, which we found adhered to the desired output JSON schema 100% of the time and never chose to return `null` for `readable_name`. We experimented with both `gpt-3.5-turbo` and `gpt-4` for AutoDoc and found both resulted in comparable synthesis performance on REGEX. However, GPT-4 was significantly slower: whereas `gpt-3.5-turbo` averaged 10-20 seconds for one iteration of AutoDoc, `gpt-4` averaged upwards of 2 minutes per iteration. We therefore chose to use `gpt-3.5-turbo` in the experiments reported in §4.

Unlike for the LLM Solver, we do not provide any few-shot examples of the desired transformations; all of this behavior is *zero-shot*, making AutoDoc an extremely domain-general technique that is easy to implement across a variety of settings.

### A.3  TASK-EXAMPLE SELECTION METHODS

Our LLM-guided program synthesis method (Eq. 3) requires selecting a set of few-shot examples for prompting. As the set of solved tasks grows, the set of possible examples exceeds the size of the LLM context window. This issue particularly affects non-compressive methods, such as the LLM Solver baseline. However, even with program compression—which substantially reduces the length of the program examples—LILO still requires subsampling from the total set of possible examples.

We experimented with two different methods for task example selection: a naive random sampling method and a task-example selection method (Liu et al., 2022) based on cosine similarity between the task descriptions of the example $\vec{d_x}$ and the target $\vec{d_t}$:

$$\text{score}(\vec{d_x}, \vec{d_t}) = \frac{\vec{d_x} \cdot \vec{d_t}}{\|\vec{d_x}\|\|\vec{d_t}\|}$$

In our implementation, we used embeddings from `text-embedding-ada-002` via the OpenAI API to pre-compute pairwise similarities between all task descriptions in each domain. For both selection methods, we construct the prompt dynamically to fit as many examples as possible.

We ran a head-to-head comparison between the two sampling methods for our main LILO model. As Fig. 7 and Tab. 2 show, we did not observe a significant improvement from the cosine similarity example selection method, though introducing determinism did have the effect of reducing the variance across runs in the REGEX domain. In absence of evidence justifying additional methods complexity, we chose to use random sampling for the results reported in §4.

It is possible that the use of compression in LILO reduces the need for targeted example selection, since we are able to fit approx. 20-40 examples per prompt across all domains. We also noted a tendency for the cosine similarity sampling to be oversensitive to superficial lexical overlap in the task descriptions; e.g., two tasks might involve very different programs but both include the word "six" as an argument, resulting in high cosine similarity. Thus, methods that explicitly finetune a model to infer similarity between (observed) example and (unobserved) target *programs* (i.e., Target Similarity Tuning from Poesia et al., 2022) could offer clearer performance advantages.

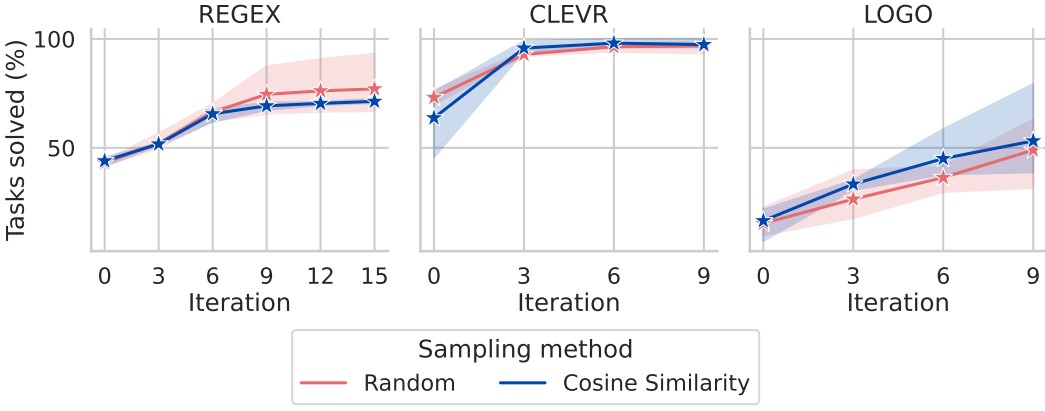

Figure 7: Head-to-head comparison between task example selection methods for the main LILO model.

| | TASKS SOLVED (%) | | | | | | | | |
| | REGEX | | | CLEVR | | | LOGO | | |
| MODEL | max | mean | std | max | mean | std | max | mean | std |
|---|---|---|---|---|---|---|---|---|---|
| LILO (Random) | 93.20 | 77.07 | 14.14 | 99.03 | 96.76 | 3.12 | 73.87 | 48.95 | 22.15 |
| LILO (Similarity) | 72.60 | 71.33 | 1.10 | 100.00 | 97.41 | 2.24 | 79.28 | 53.15 | 22.67 |

Table 2: Final performance of task example selection methods for the main LILO model.

## A.4 IMPLEMENTATION DETAILS

We provide a brief summary of key implementation details relevant to the experiments that are not reported in §4. We ran all experiments on AWS EC2 instances with machine specs tailored to suit the computational workload of each experiment.

**Enumerative search.** For experiments involving enumerative search, which is an embarrassingly parallel workload that scales linearly with the number of available CPUs, we ran on 96-CPU `c5.24xlarge` instances. These machines have the highest CPU count in the `c5` machine class. To take maximal advantage of the CPU parallelism, we set `batch_size=96` for these experiments (i.e., each iteration searches for solutions for a subset of 96 tasks). A convenient consequence of this implementation choice is that each task is allocated to a single, dedicated CPU, so the overall wall clock runtime of a single search iteration is equal to the per-task enumeration time budget. We set the enumeration budget on a per-domain basis using the timeouts from Wong et al. (2021) (REGEX = 1000s, CLEVR = 600s, LOGO = 1800s). We ran DreamCoder until convergence on all domains. For CLEVR and LOGO, we performed 10 iterations of search, while for REGEX, we observed that the solve rate was still increasing at iteration 10, so we used a higher search budget of 16 iterations for this domain. Following Wong et al. (2021) and based on a common practice in machine learning, we limited evaluation of the test set to every 3 iterations due to the computational cost of enumerative search.

**GPT language models.** For experiments in which GPT LLMs perform program search, the bulk of the computational workload is effectively offloaded to OpenAI's servers. Locally, the only requirements are that our machine is able to make API queries, process the results, and run compression. Accordingly, these experiments are run on `c5.2xlarge` machines with 8 CPUs each. (For experiments involving combinations of GPT queries and DreamCoder search, we use the larger `c5.24xlarge` machines.) To ensure comparability in solver performance between LLM-based and enumerative search-based experiments, we also run the LLM experiments with `batch_size=96` so that the learning timelines are aligned.

**Stitch.** For compression, we make use of the Stitch Python bindings, which interface with a fast backend written in Rust (https://stitch-bindings.readthedocs.io/en/stable/). Stitch exposes various hyperparameters, the most important of which are `iterations`, which governs the number of abstractions produced, and `max-arity`, which governs the maximum number of arguments that each abstraction can take. For all experiments, we set these to a constant `iterations=10` and `max-arity=3`. We note that Stitch will only produce an abstraction if it is *compressive*; i.e., it appears in multiple programs, and rewriting the corpus in terms of the abstraction reduces the overall description length. For this reason, in rare cases early on in learning, when only a handful of solved programs are available, the actual library size can be smaller than `iterations`. This behavior is beneficial in that it avoids introducing abstractions that have no utility and that might potentially negatively affect performance.

A summary of hyperparameters can be found in A.5. For further implementation details, we refer to our codebase: github.com/gabegrand/lilo.

## A.5 HYPERPARAMETERS

We provide a summary of all key hyperparameters used in each component of LILO.

DREAMCODER

| | |
|---|---|
| **Batch size:** | 96 tasks |
| **Global iterations:** | 10 (CLEVR, LOGO), 16 (REGEX) |
| **Search timeouts:** | 600s (CLEVR), 1000s (REGEX), 1800s (LOGO) |
| **Neural recognition model:** | 10K training steps / iteration |

STITCH

| | |
|---|---|
| **Max iterations:** | 10 (Controls max library size) |
| **Max arity:** | 3 (Controls max arity of abstractions) |

LILO: LLM SYNTHESIZER

| | |
|---|---|
| **Prompts per task:** | 4 |
| **Samples per prompt:** | 4 |
| **GPT Model:** | `code-davinci-002` |
| **Temperature:** | 0.90 |
| **Max completion tokens $\beta$:** | 4.0x (Multiplier w/r/t the final prompt program.) |

LILO: AUTODOC

**Max usage examples:** 10
**GPT Model:** `gpt-3.5-turbo-0301` / `gpt-4-0314`
**Top-P:** 0.10
**Max completion tokens:** 256

# B    Experiments and Results

## B.1    Domain Details

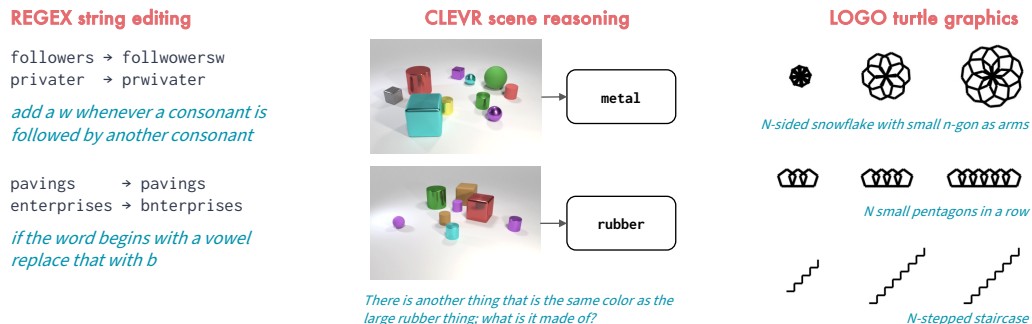

Figure 8: **Overview of domains.** We evaluate LILO on three *language-annotated* program synthesis domains: *string editing* with regular expressions, *scene reasoning* on the CLEVR dataset, and *graphics composition* in the 2D Logo turtle graphics language.

| Domain | #Tasks | | Description length | | String length | |
|--------|--------|------|------|------|------|------|
| | Train | Test | Train | Test | Train | Test |
| REGEX | 491 | 500 | $38.95 \pm 26.11$ | $41.03 \pm 27.02$ | $276.47 \pm 179.92$ | $262.74 \pm 172.69$ |
| CLEVR | 191 | 103 | $32.95 \pm 15.78$ | $30.82 \pm 15.49$ | $361.62 \pm 182.06$ | $387.44 \pm 184.19$ |
| LOGO | 200 | 111 | $24.65 \pm 8.71$ | $27.79 \pm 8.19$ | $250.98 \pm 92.75$ | $287.17 \pm 89.65$ |

Table 3: **Summary statistics for the domains used in this paper.** Description length is the number of terminals, lambda-abstractions and applications necessary to uniquely describe the ground truth program for each task; string length is the length of each program in terms of characters. Both are reported as the mean over the entire dataset plus/minus one standard deviation.

**REGEX: String editing.** We evaluate on a domain of *structured string transformation problems*–a classic task in inductive program synthesis (Lau & Weld, 1998). The dataset, originally introduced in Andreas et al. (2018), contains procedurally-generated regular expressions that implement transformations on strings (e.g., *if the word ends with a consonant followed by "s", replace that with b*). Task examples consist of input/output pairs where the inputs are strings randomly sampled from an English dictionary and the outputs are the result of applying a particular string transformation. Following prior work (Ellis et al., 2021; Wong et al., 2021), the base DSL in this domain contains functional various programming primitives for string manipulation (map, fold, cons, car, cdr, length, index) and character constants. Each example comes with a synthetic language description of the task, which was generated by template based on human annotations (Andreas et al., 2018).

**CLEVR: Scene reasoning.** We extend our approach to a *visual question answering* (VQA) task based on the CLEVR dataset (Johnson et al., 2017). Following successful efforts in modeling VQA as program synthesis (Andreas et al., 2016; Hu et al., 2017), each synthesis task is specified by a structured input scene and a natural language question. Outputs can be one of several types, including a number (*how many red rubber things are there?*), a boolean value (*are there more blue things than green?*), or another scene (*what if all of the red things turned blue?*). The dataset, designed by Wong et al. (2021), uses a modified subset of the original CLEVR tasks and introduces new task types that require imagining or generating new scenes (e.g., *how many metal things would be left if all the blue cylinders were removed?*) that require learning new abstractions. The base DSL includes functional programming primitives similar to the regular expression domain, with domain-specific query functions and constants (e.g., get_color(x); get_shape(x); blue; cube). Input scenes are specified *symbolically* as scene graphs consisting of an array of structured objects defined as a dictionary of their attributes, and programs are designed to manipulate these structured arrays. Synthetic language annotations were generated based on the original high-level templates in Johnson et al. (2017) and human annotations were collected by Wong et al. (2021).

**LOGO: Turtle graphics.** Following in a long tradition of modeling *vision as inverse graphics*, (Knill & Richards, 1996; Kersten et al., 2004; Yuille & Kersten, 2006; Lee & Mumford, 2003; Wu et al., 2015; Yildirim et al., 2020; Wu et al., 2017; Yi et al., 2018; Gothoskar et al., 2021) we evaluate on a domain of *compositional drawing problems*. The dataset, originally introduced in (Wong et al., 2021) and based on a simpler dataset from (Ellis et al., 2021), contains programs that generate shapes and designs in a vector graphics language. The DSL is based on Logo Turtle graphics (Abelson & diSessa, 1986), which originated from early symbolic AI research. Program expressions control the movement and direction of a pen (classically represented as a Turtle) on a canvas and can involve complex symmetries and recursions (e.g., *a seven sided snowflake with a short line and a small triangle as arms; a small triangle connected by a big space from a small circle*). The base DSL includes for loops, a stack for saving/restoring the pen state, and arithmetic on angles and distances (Ellis et al., 2021). Synthetic language annotations were generated with high-level templates over the objects and relations in each task; human annotations were collected by Wong et al. (2021).

## B.2 Learned Libraries and Graphical Maps

We generated graphical visualizations of the libraries learned by the best Lilo model for each domain. Each graph includes the DSL primitives, the learned and named abstractions, and a random sample of 3 solved tasks that invoke each abstraction. Arrows indicate direction of reference; i.e., `fn_1 -> fn_2` indicates that `fn_1` invokes `fn_2`, and analogously for the tasks.

### B.2.1 LIBRARY FOR REGEX

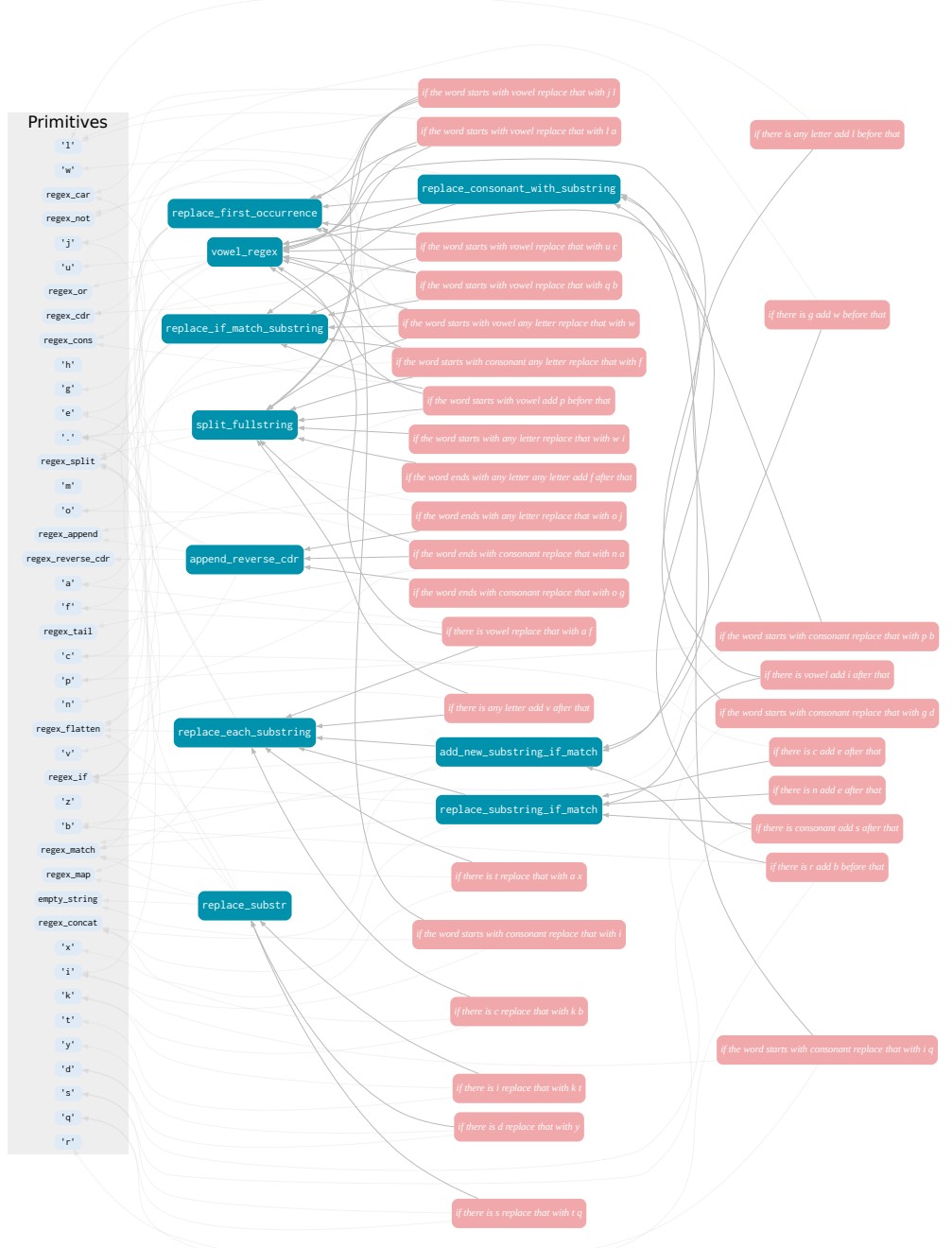

Figure 9: **Graphical map of REGEX library learned by LILO.** Named abstractions (turquoise) are hierarchically composed of other abstractions and ground out in the base DSL primitives (gray box).

```
(fn_42) vowel_regex :: tsubstr
(regex_or 'a' (regex_or 'e' (regex_or 'i' (regex_or 'o' 'u'))))
{- Regular expression that matches any vowel ('a', 'e', 'i', 'o', 'u'). Used in various
functions to identify and modify words based on vowel presence and position. -}

{- Example usages -}
--if there is consonant add s after that
(λ (replace_substring_if_match 's' (regex_not vowel_regex) $0))
--if the word starts with vowel replace that with j l
(λ (regex_if (regex_match (regex_not vowel_regex) (regex_car (split_fullstring $0))) $0
(replace_first_occurrence $0 'l' 'j')))
--if the word starts with vowel replace that with u c
(λ (replace_if_match_substring $0 (replace_first_occurrence $0 'c' 'u') vowel_regex))

(fn_43) replace_substr :: tfullstr -> tsubstr -> tsubstr -> tfullstr
(λ (λ (λ (regex_flatten (regex_map (λ (regex_if (regex_match $1 $0) $2 $0))
(regex_split empty_string $2))))))
{- Replaces all instances of a given substring $1 in a full string $0 with another
substring $2. The substrings are separated by empty spaces. -}

{- Example usages -}
--if there is d replace that with y
(λ (replace_substr $0 'y' 'd'))
--if there is i replace that with k t
(λ (replace_substr $0 (regex_concat 'k' 't') 'i'))
--if there is s replace that with t q
(λ (replace_substr $0 (regex_concat 't' 'q') 's'))

(fn_44) replace_first_occurrence :: tfullstr -> tsubstr -> tsubstr -> tfullstr
(λ (λ (λ (regex_flatten (regex_cons $0 (regex_cons $1 (regex_cdr (regex_split '.'
$2)))))))))
{- Replaces the first occurrence of a substring $1 in a full string $0 with another
substring $2. The substrings are separated by periods. -}

{- Example usages -}
--if the word starts with vowel replace that with q b
(λ (replace_if_match_substring $0 (replace_first_occurrence $0 'b' 'q') vowel_regex))
--if the word starts with consonant replace that with i
(λ (replace_first_occurrence $0 empty_string 'i'))
--if the word starts with vowel replace that with l a
(λ (regex_if (regex_match (regex_not vowel_regex) (regex_car (split_fullstring $0))) $0
(replace_first_occurrence $0 'a' 'l')))

(fn_45) replace_each_substring :: tfullstr -> (tsubstr -> tsubstr) -> tfullstr
(λ (λ (regex_flatten (regex_map $0 (regex_split '.' $1)))))
{- Replaces each substring separated by periods in a given full string with a new
substring. The new substring can be manipulated with a λ function that takes each
substring as input. -}

{- Example usages -}
--if there is t replace that with a x
(λ (replace_each_substring $0 (λ (regex_if (regex_match 't' $0) (regex_concat 'a' 'x')
$0))))
--if there is vowel replace that with a f
(λ (replace_each_substring $0 (λ (regex_if (regex_match vowel_regex $0) (regex_concat
'a' 'f') $0))))
--if there is c replace that with k b
(λ (replace_each_substring $0 (λ (regex_if (regex_match 'c' $0) (regex_concat 'k' 'b')
$0))))

(fn_46) replace_if_match_substring :: tfullstr -> tfullstr -> tsubstr -> tfullstr
(λ (λ (λ (regex_if (regex_match $0 (regex_car (regex_split '.' $2))) $1 $2))))
{- Replaces a given substring $2 in a full string $0 with another substring $1 if the
beginning of the string matches the target substring. All substrings are separated by
periods. -}
```

```
{- Example usages -}
--if the word starts with vowel add p before that
(λ (replace_if_match_substring $0 (regex_flatten (regex_cons 'p' (split_fullstring
$0))) vowel_regex))
--if the word starts with consonant any letter replace that with f
(λ (replace_if_match_substring $0 (regex_flatten (regex_cons 'f' (regex_cdr (regex_cdr
(split_fullstring $0))))) (regex_not vowel_regex)))
--if the word starts with vowel any letter replace that with w
(λ (replace_if_match_substring $0 (regex_flatten (regex_cons 'w' (regex_cdr (regex_cdr
(split_fullstring $0))))) vowel_regex))

(fn_47) add_new_substring_if_match :: tsubstr -> tsubstr -> tfullstr -> tfullstr
(λ (λ (λ (replace_each_substring $0 (λ (regex_if (regex_match $2 $0) (regex_concat $3
$0) $0))))))
{- Replaces each substring separated by periods in a given full string with a new
substring, if a specified substring is found. The new substring can be manipulated with
a λ function that takes each substring as input. -}

{- Example usages -}
--if there is g add w before that
(λ (add_new_substring_if_match 'w' 'g' $0))
--if there is any letter add l before that
(λ (add_new_substring_if_match 'l' '.' $0))
--if there is r add b before that
(λ (add_new_substring_if_match 'b' 'r' $0))

(fn_48) append_reverse_cdr :: tfullstr -> tsubstr -> tfullstr
(λ (λ (regex_flatten (regex_append $0 (regex_reverse_cdr (regex_split '.' $1))))))
{- Appends a new substring to the end of the given full string and reverses the order
of all substrings except for the last one (which is removed). -}

{- Example usages -}
--if the word ends with consonant replace that with o g
(λ (append_reverse_cdr $0 (regex_concat 'o' 'g')))
--if the word ends with consonant replace that with n a
(λ (regex_if (regex_match 'e' (regex_tail (split_fullstring $0))) $0
(append_reverse_cdr $0 (regex_concat 'n' 'a'))))
--if the word ends with any letter replace that with o j
(λ (append_reverse_cdr $0 (regex_concat 'o' 'j')))

(fn_49) replace_substring_if_match :: tsubstr -> tsubstr -> tfullstr -> tfullstr
(λ (λ (λ (replace_each_substring $0 (λ (regex_if (regex_match $2 $0) (regex_concat $0
$3) $0))))))
{- Replaces each substring separated by periods in a given full string with a new
substring, if a specified substring is found, using a λ function that takes the current
substring as input and replaces it with a new substring based on a condition. -}

{- Example usages -}
--if there is vowel add i after that
(λ (replace_substring_if_match 'i' vowel_regex $0))
--if there is c add e after that
(λ (replace_substring_if_match 'e' 'c' $0))
--if there is n add e after that
(λ (replace_substring_if_match 'e' 'n' $0))

(fn_50) split_fullstring :: tfullstr -> list(tsubstr)
(λ (regex_split '.' $0))
{- Splits a given full string into a list of substrings separated by periods. -}

{- Example usages -}
--if the word ends with any letter any letter add f after that
(λ (regex_flatten (regex_append (regex_concat 'f' empty_string) (split_fullstring
$0))))
```

```
--if the word starts with any letter replace that with w i
(λ (regex_flatten (regex_cons (regex_concat 'w' 'i') (regex_cdr (split_fullstring
$0)))))
--if there is any letter add v after that
(λ (replace_each_substring $0 (λ (regex_tail (regex_map (λ (regex_concat $1 'v'))
(split_fullstring $1))))))

(fn_51) replace_consonant_with_substring :: tsubstr -> tsubstr -> tfullstr -> tfullstr
(λ (λ (λ (replace_if_match_substring $0 (replace_first_occurrence $0 $1 $2) (regex_not
vowel_regex)))))
{- Replaces the first occurrence of a consonant at the beginning of a given full string
with a specified substring. The target substring can also be modified before
replacement using another specified substring. -}

{- Example usages -}
--if the word starts with consonant replace that with i q
(λ (replace_consonant_with_substring 'i' 'q' $0))
--if the word starts with consonant replace that with g d
(λ (replace_consonant_with_substring 'g' 'd' $0))
--if the word starts with consonant replace that with p b
(λ (replace_consonant_with_substring 'p' 'b' $0))
```

## B.2.2 LIBRARY FOR CLEVR

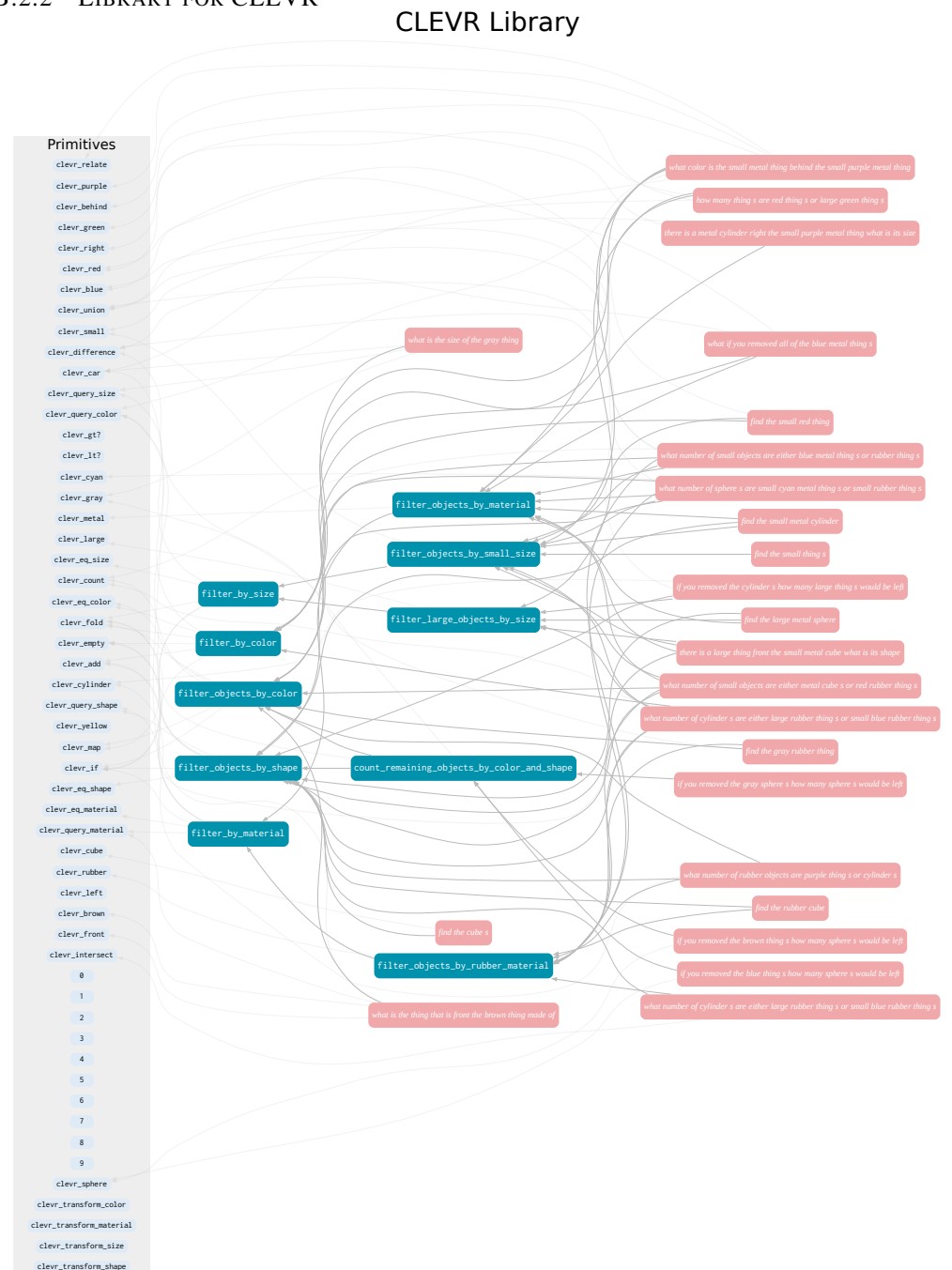

Figure 10: **Graphical map of CLEVR library learned by LILO.** Named abstractions (turquoise) are hierarchically composed of other abstractions and ground out in the base DSL primitives (gray box).

```
(fn_54) filter_by_size :: tclevrsize -> list(tclevrobject) -> list(tclevrobject)
(λ (λ (clevr_fold $0 $0 (λ (λ (clevr_map (λ (clevr_if (clevr_eq_size (clevr_query_size
$0) $4) $0 $2)) $0))))))
{- Returns a list of objects in the input list that have the specified size. -}

(fn_55) filter_by_color :: tclevrcolor -> list(tclevrobject) -> list(tclevrobject)
(λ (λ (clevr_fold $0 clevr_empty (λ (λ (clevr_if (clevr_eq_color (clevr_query_color
$1) $3) (clevr_add $1 $0) $0))))))
{- Returns a list of objects in the input list that have the specified color. -}

{- Example usages -}
--what color is the small metal thing behind the small purple metal thing
(λ (clevr_query_color (clevr_car (filter_objects_by_material
(filter_objects_by_small_size (clevr_relate (clevr_car (filter_by_color clevr_purple
(filter_objects_by_material (filter_objects_by_small_size $0)))) clevr_behind $0))))))
--what is the size of the gray thing
(λ (clevr_query_size (clevr_car (filter_by_color clevr_gray $0))))
--how many thing s are red thing s or large green thing s
(λ (clevr_count (clevr_union (filter_by_color clevr_red $0)
(filter_large_objects_by_size (filter_by_color clevr_green $0)))))

(fn_56) filter_by_material :: tclevrmaterial -> list(tclevrobject) ->
list(tclevrobject)
(λ (λ (clevr_fold $0 clevr_empty (λ (λ (clevr_if (clevr_eq_material
(clevr_query_material $1) $3) (clevr_add $1 $0) $0))))))
{- Returns a list of objects in the input list that have the specified material. -}

(fn_57) filter_objects_by_shape :: tclevrshape -> list(tclevrobject) ->
list(tclevrobject)
(λ (λ (clevr_fold $0 clevr_empty (λ (λ (clevr_if (clevr_eq_shape (clevr_query_shape
$1) $3) (clevr_add $1 $0) $0))))))
{- Filters a list of objects to include only those with the specified shape. -}

{- Example usages -}
--find the cube s
(λ (filter_objects_by_shape clevr_cube $0))
--find the rubber cube
(λ (filter_objects_by_rubber_material (filter_objects_by_shape clevr_cube $0)))
--if you removed the cylinder s how many large thing s would be left
(λ (clevr_count (clevr_difference (filter_large_objects_by_size $0)
(filter_objects_by_shape clevr_cylinder $0))))

(fn_58) filter_objects_by_color :: tclevrcolor -> list(tclevrobject) ->
list(tclevrobject)
(λ (λ (clevr_fold $0 $0 (λ (λ (clevr_map (λ (clevr_if (clevr_eq_color
(clevr_query_color $0) $4) $0 $2)) $0))))))
{- Returns a list of objects in the input list that have the specified color. -}

{- Example usages -}
--find the gray rubber thing
(λ (filter_objects_by_rubber_material (filter_objects_by_color clevr_gray $0)))
--what is the thing that is front the brown thing made of
(λ (clevr_query_material (clevr_car (clevr_relate (clevr_car (filter_objects_by_color
clevr_brown $0)) clevr_front $0))))
--what number of small objects are either metal cube s or red rubber thing s
(λ (clevr_count (filter_objects_by_small_size (clevr_union (filter_objects_by_material
(filter_objects_by_shape clevr_cube $0)) (filter_objects_by_rubber_material
(filter_objects_by_color clevr_red $0))))))

(fn_59) filter_objects_by_small_size :: list(tclevrobject) -> list(tclevrobject)
(λ (filter_by_size clevr_small $0))
{- Returns a list of objects in the input list that are small in size. -}

{- Example usages -}
```

```
--find the small red thing
(λ (filter_objects_by_small_size (filter_objects_by_color clevr_red $0)))
--find the small thing s
(λ (filter_objects_by_small_size $0))
--what number of small objects are either blue metal thing s or rubber thing s
(λ (clevr_count (filter_objects_by_small_size (clevr_union
(filter_objects_by_rubber_material $0) (filter_objects_by_material
(filter_objects_by_color clevr_blue $0))))))

(fn_60) filter_objects_by_material :: list(tclevrobject) -> list(tclevrobject)
(λ (filter_by_material clevr_metal $0))
{- Returns a list of objects in the input list that have the specified material. -}

{- Example usages -}
--there is a metal cylinder right the small purple metal thing what is its size
(λ (clevr_if (clevr_eq_shape clevr_cube (clevr_query_shape (clevr_car (clevr_relate
(clevr_car (clevr_union $0 (filter_objects_by_material $0))) clevr_right $0))))
clevr_small clevr_large))
--what if you removed all of the blue metal thing s
(λ (clevr_difference $0 (filter_objects_by_color clevr_blue (filter_objects_by_material
$0))))
--find the small metal cylinder
(λ (filter_objects_by_small_size (filter_objects_by_material (filter_objects_by_shape
clevr_cylinder $0))))

(fn_61) count_remaining_objects_by_color_and_shape :: list(tclevrobject) -> tclevrcolor
-> tclevrshape -> int
(λ (λ (λ (clevr_count (clevr_difference (filter_objects_by_shape $0 $2)
(filter_objects_by_color $1 $2))))))
{- Counts the number of objects that remain after removing objects of a specified color
and shape from the input list of objects. -}

{- Example usages -}
--if you removed the brown thing s how many sphere s would be left
(λ (count_remaining_objects_by_color_and_shape $0 clevr_brown clevr_sphere))
--if you removed the red cube s how many cube s would be left
(λ (count_remaining_objects_by_color_and_shape $0 clevr_red clevr_cube))
--if you removed the cyan cylinder s how many cylinder s would be left
(λ (count_remaining_objects_by_color_and_shape $0 clevr_cyan clevr_cylinder))

(fn_62) filter_objects_by_rubber_material :: list(tclevrobject) -> list(tclevrobject)
(λ (filter_by_material clevr_rubber $0))
{- Returns a list of objects in the input list that have rubber as their material. -}

{- Example usages -}
--what number of sphere s are small cyan metal thing s or small rubber thing s
(λ (clevr_count (clevr_union (filter_objects_by_material (filter_objects_by_small_size
(filter_by_color clevr_cyan (filter_objects_by_shape clevr_sphere $0))))
(filter_objects_by_rubber_material (filter_objects_by_small_size
(filter_objects_by_shape clevr_sphere $0))))))
--what number of rubber objects are purple thing s or cylinder s
(λ (clevr_count (filter_objects_by_rubber_material (clevr_union
(filter_objects_by_shape clevr_cylinder $0) (filter_objects_by_color clevr_purple
$0)))))
--what number of cylinder s are either large rubber thing s or small blue rubber thing s
(λ (clevr_count (clevr_intersect (filter_objects_by_rubber_material $0)
(filter_objects_by_shape clevr_cylinder $0))))

(fn_63) filter_large_objects_by_size :: list(tclevrobject) -> list(tclevrobject)
(λ (filter_by_size clevr_large $0))
{- Returns a list of objects in the input list that are large in size. -}

{- Example usages -}
--find the large metal sphere
```

```
(λ (filter_large_objects_by_size (filter_objects_by_material (filter_objects_by_shape
clevr_sphere $0))))
--there is a large thing front the small metal cube what is its shape
(λ (clevr_query_shape (clevr_car (filter_large_objects_by_size (clevr_relate (clevr_car
(filter_objects_by_small_size (filter_objects_by_material (filter_objects_by_shape
clevr_cube $0)))) clevr_front $0)))))
--what number of cylinder s are either large rubber thing s or small blue rubber thing s
(λ (clevr_count (filter_objects_by_shape clevr_cylinder (clevr_union
(filter_objects_by_rubber_material (filter_large_objects_by_size $0))
(filter_objects_by_small_size (filter_by_color clevr_blue
(filter_objects_by_rubber_material $0)))))))
```

### B.2.3  LIBRARY FOR LOGO

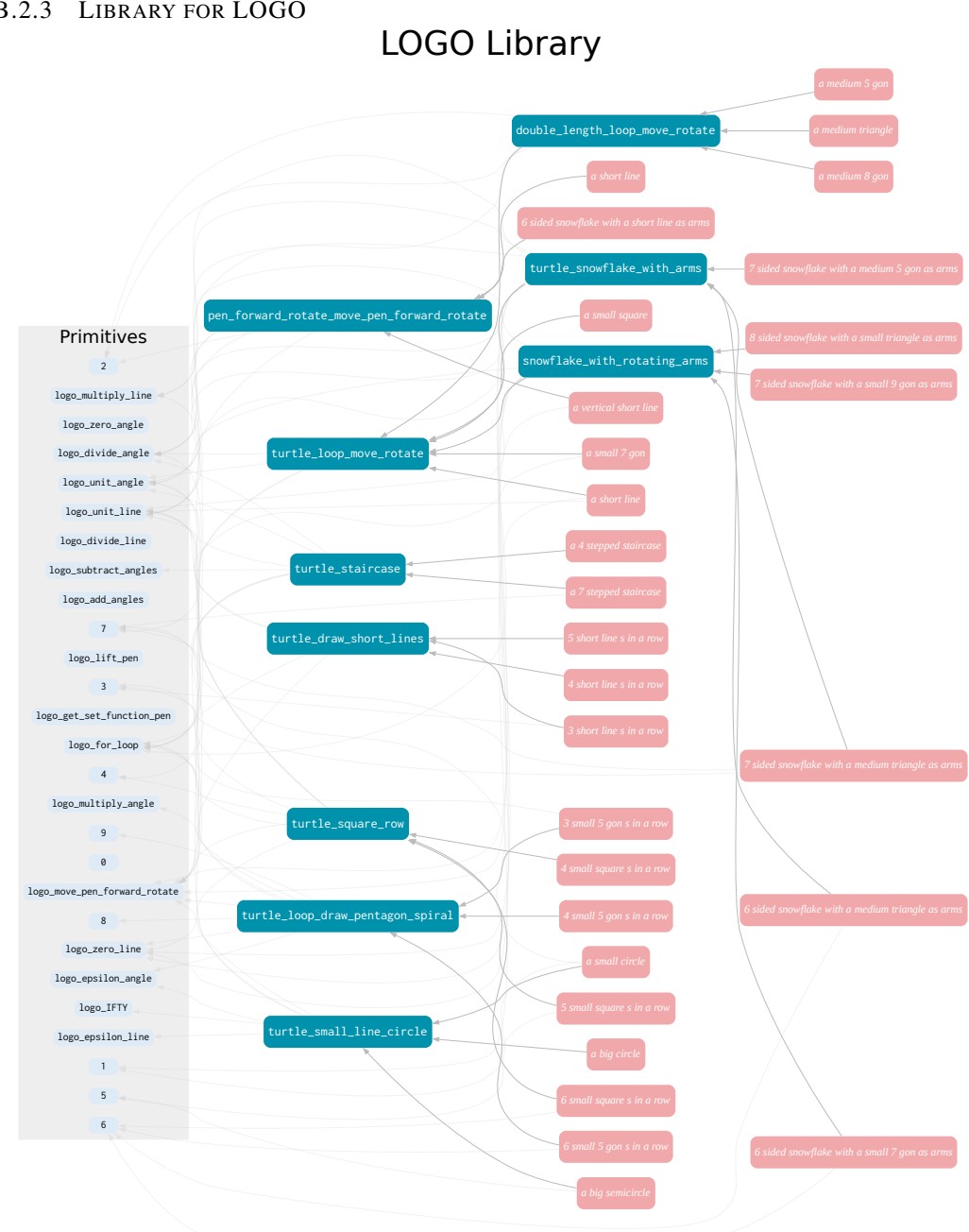

Figure 11: **Graphical map of LOGO library learned by LILO.** Named abstractions (turquoise) are hierarchically composed of other abstractions and ground out in the base DSL primitives (gray box).

```
(fn_27) turtle_loop_move_rotate :: turtle -> int -> tlength -> turtle
(λ (λ (λ (logo_for_loop $1 (λ (λ (logo_move_pen_forward_rotate $2 (logo_divide_angle
logo_unit_angle $3) $0))) $2)))))
{- Repeatedly move the turtle forward and rotate it by a specified angle, creating a
loop of a specific number of sides with a given line length. -}

{- Example usages -}
--a small square
(λ (turtle_loop_move_rotate $0 4 logo_unit_line))
--a small 7 gon
(λ (turtle_loop_move_rotate $0 7 logo_unit_line))
--a short line
(λ (turtle_loop_move_rotate $0 1 logo_unit_line))

(fn_28) turtle_staircase :: turtle -> int -> turtle
(λ (λ (logo_for_loop $0 (λ (λ (logo_move_pen_forward_rotate logo_unit_line
(logo_divide_angle logo_unit_angle 4) (logo_move_pen_forward_rotate logo_unit_line
(logo_subtract_angles logo_unit_angle (logo_divide_angle logo_unit_angle 4)) $0))))
$1)))
{- Creates a staircase pattern by repeatedly moving the turtle forward and rotating it
at a specific angle. The number of steps in the staircase is determined by the function
argument. -}

{- Example usages -}
--a 4 stepped staircase
(λ (turtle_staircase $0 4))
--a 7 stepped staircase
(λ (turtle_staircase $0 7))
--a 4 stepped staircase
(λ (turtle_staircase $0 4))

(fn_29) turtle_loop_draw_pentagon_spiral :: turtle -> int -> turtle
(λ (λ (logo_for_loop $0 (λ (λ (logo_move_pen_forward_rotate logo_zero_line
(logo_multiply_angle logo_epsilon_angle 8) (logo_for_loop 9 (λ (λ
(logo_move_pen_forward_rotate logo_unit_line (logo_multiply_angle logo_epsilon_angle 8)
$0))) $0)))) $1)))
{- Creates a spiral of pentagons by repeatedly drawing a pentagon and incrementing the
angle of each side on each iteration. The number of pentagons in the spiral is
determined by the function argument. -}

{- Example usages -}
--4 small 5 gon s in a row
(λ (turtle_loop_draw_pentagon_spiral $0 4))
--3 small 5 gon s in a row
(λ (turtle_loop_draw_pentagon_spiral $0 3))
--6 small 5 gon s in a row
(λ (turtle_loop_draw_pentagon_spiral $0 6))

(fn_30) turtle_square_row :: turtle -> int -> turtle
(λ (λ (logo_for_loop $0 (λ (λ (logo_move_pen_forward_rotate logo_zero_line
(logo_divide_angle logo_unit_angle 4) (logo_for_loop 7 (λ (λ
(logo_move_pen_forward_rotate logo_unit_line (logo_divide_angle logo_unit_angle 4)
$0))) $0))))) $1)))
{- Draws a row of small squares using repeated forward motion and rotation. The number
of squares in the row is determined by the function argument. -}

{- Example usages -}
--4 small square s in a row
(λ (turtle_square_row $0 4))
--6 small square s in a row
(λ (turtle_square_row $0 6))
--5 small square s in a row
(λ (turtle_square_row $0 5))
```

```
(fn_31) turtle_snowflake_with_arms :: turtle -> int -> int -> turtle
(λ (λ (λ (logo_for_loop $0 (λ (λ (turtle_loop_move_rotate
(logo_move_pen_forward_rotate logo_zero_line (logo_divide_angle logo_unit_angle $2) $0)
$3 (logo_multiply_line logo_unit_line 2)))) $2))))
{- Draws a snowflake shape with given number of arms, each made up of a line of specified
length that is rotated at a specific angle. The angle by which the lines are rotated
increases with each iteration of the loop, creating an intricate snowflake pattern. -}

{- Example usages -}
--7 sided snowflake with a medium 5 gon as arms
(λ (turtle_snowflake_with_arms $0 5 7))
--6 sided snowflake with a medium triangle as arms
(λ (turtle_snowflake_with_arms $0 3 6))
--7 sided snowflake with a medium triangle as arms
(λ (turtle_snowflake_with_arms $0 3 7))

(fn_32) turtle_small_line_circle :: turtle -> int -> turtle
(λ (λ (logo_for_loop logo_IFTY (λ (λ (logo_move_pen_forward_rotate (logo_multiply_line
logo_epsilon_line $2) logo_epsilon_angle $0))) $1)))
{- Moves the turtle forward and rotates it repeatedly to draw a small circle with a
given line length. The number of iterations is determined by the function argument. -}

{- Example usages -}
--a small circle
(λ (logo_for_loop 7 (λ (λ (turtle_small_line_circle $0 1))) $0))
--a big semicircle
(λ (turtle_small_line_circle $0 5))
--a big circle
(λ (logo_for_loop 7 (λ (λ (turtle_small_line_circle $0 5))) $0))

(fn_33) snowflake_with_rotating_arms :: turtle -> int -> int -> turtle
(λ (λ (λ (logo_for_loop $0 (λ (λ (turtle_loop_move_rotate
(logo_move_pen_forward_rotate logo_zero_line (logo_divide_angle logo_unit_angle $2) $0)
$3 logo_unit_line))) $2))))
{- Draws a snowflake shape with given number of arms, each made up of a line of specified
length that is rotated at a specific angle. The angle by which the lines are rotated
increases with each iteration of the loop, creating an intricate snowflake pattern. -}

{- Example usages -}
--7 sided snowflake with a small 9 gon as arms
(λ (snowflake_with_rotating_arms $0 9 7))
--6 sided snowflake with a small 7 gon as arms
(λ (snowflake_with_rotating_arms $0 7 6))
--8 sided snowflake with a small triangle as arms
(λ (snowflake_with_rotating_arms $0 3 8))

(fn_34) double_length_loop_move_rotate :: int -> turtle -> turtle
(λ (λ (turtle_loop_move_rotate $0 $1 (logo_multiply_line logo_unit_line 2))))
{- Moves and rotates the turtle in a loop, with each iteration doubling the length of
the turtle's movement. -}

{- Example usages -}
--a medium 5 gon
(λ (double_length_loop_move_rotate 5 $0))
--a medium triangle
(λ (double_length_loop_move_rotate 3 $0))
--a medium 8 gon
(λ (double_length_loop_move_rotate 8 $0))

(fn_35) turtle_draw_short_lines :: turtle -> int -> turtle
(λ (λ (logo_for_loop $0 (λ (λ (logo_move_pen_forward_rotate logo_unit_line
logo_unit_angle $0))) $1)))
{- Draws a specified number of short lines in a row using repeated forward motion and
rotation. -}
```

```
{- Example usages -}
--5 short line s in a row
(λ (turtle_draw_short_lines $0 5))
--4 short line s in a row
(λ (turtle_draw_short_lines $0 4))
--3 short line s in a row
(λ (turtle_draw_short_lines $0 3))

(fn_36) pen_forward_rotate_move_pen_forward_rotate :: turtle -> int -> tlength -> turtle
(λ (λ (λ (logo_move_pen_forward_rotate $0 (logo_divide_angle logo_unit_angle $1)
(logo_move_pen_forward_rotate logo_unit_line (logo_divide_angle logo_unit_angle 2)
$2)))))
{- Moves the turtle forward and rotates it at a given angle. Then moves the turtle
forward again and rotates it at half the angle, creating a pivot point for the turtle
to change direction. The distance the turtle moves each time is determined by a given
length parameter. -}

{- Example usages -}
--a vertical short line
(λ (pen_forward_rotate_move_pen_forward_rotate $0 4 logo_zero_line))
--a short line
(λ (pen_forward_rotate_move_pen_forward_rotate $0 2 logo_unit_line))
--6 sided snowflake with a short line as arms
(λ (logo_for_loop 7 (λ (λ (pen_forward_rotate_move_pen_forward_rotate $0 3
logo_unit_line))) $0))
```

## B.3 BENCHMARK COMPARISON TO PRIOR WORK

| Language | Model | Strings ($n_{test}$ = 500) | Graphics ($n_{test}$ = 111) | | Scenes ($n_{test}$ = 115) | |
|---|---|---|---|---|---|---|
| | | % Solved | % Solved (Best) | % Solved (Mean) | % Solved (Curric.) | % Solved (Mean.) |
| Synth train/test | DreamCoder (no language) | 33.4 | 49.55 | 42. 64 | 67.80 | 73.9 |
| Synth train/test | Multimodal (no generative translation model) | 46.00 | 26.12 | 23.20 | 76.50 | 49.5 |
| Synth train/test | LAPS in neural search | 52.20 | 92.79 | 52.93 | 95.6 | 88.1 |
| Synth train/test | LAPS + mutual exclusivity | **57.00** | 86.49 | 80.18 | **96.5** | 82.3 |
| Synth train/test | LAPS + ME + language-program compression | 54.60 | **98.19** | **81.98** | 95.6 | **95.9** |
| Synth train/human test | LAPS + ME + language-program compression | 54.60 | 89.20 | – | 97.4 | – |
| Human train/human test | LAPS + ME + language-program compression | 48.60 | 58.55 | – | 95.6 | – |
| **No language at test** | | | | | | |
| No language on train/test | Original DSL; Enumerative | 0.06 | 0.00 | – | 27.8 | – |
| No language on train/test | DreamCoder (best library): Enumerative | 27.2 | 41.44 | – | 53.6 | – |
| No lang at test | LAPS (best library): Enumerative | 33.2 | 62.16 | – | 93.04 | – |
| No lang at test | LAPS (best library): example-only neural synthesis | **52.4** | **91.0** | – | 95.6 | – |

Table 4: **Percent held-out test-tasks solved for LAPS.** *Best* reports the best model across replications; *Mean* averages across replications. (Reproduced from Wong et al., 2021.)

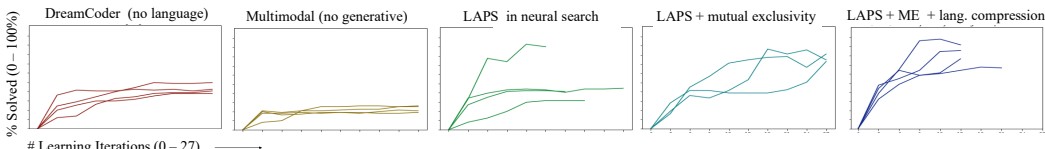

Figure 12: Learning curves comparing baselines and LAPS models in Table 4, showing % heldout tasks solved on the graphics domain over random training task orderings. (Reproduced from Wong et al., 2021.)

Our results from §4 are directly comparable to those from Wong et al. (2021). The primary results from that work are reproduced in Tab. 4, where Strings corresponds to REGEX, Graphics corresponds to LOGO, and Scenes corresponds to CLEVR. The DreamCoder baseline from our work, which uses the language-conditioned recognition model from Wong et al., is comparable to the "LAPS in neural search" condition in Tab. 4, with the key difference being that we do not use the IBM translation model component. (We also run on larger batch sizes to take full advantage of the available CPU parallelism on our cloud hardware.)

On REGEX (Strings), with the use of LLMs for search, our LLM Solver and LILO conditions perform significantly better (93.20 best vs. 57.00 best) than this prior work, even without explicitly computing language/program alignment via a translation model. On CLEVR (Scenes), our models perform comparably to LAPS: the DreamCoder baseline already solves almost all of the tasks in the test set (97.09 best), and LILO brings the best solve rate up to 99.03.

Finally, on LOGO (Graphics), our models generally underperform with respect to the results reported in LAPS (73.87 LILO best vs. 92.79 LAPS best). It is worth noting that the best run from LAPS on this domain appears to be an outlier (see Fig. 12, *LAPS in neural search*), so a comparison of average results (48.95 LILO mean vs. 52.93 LAPS mean) may be more appropriate. Moreover, even matching the 1800s search time, we were unable to obtain a DreamCoder run that matches their equivalent LAPS baseline on this domain (28.53 DreamCoder (ours) vs. 42.64 DreamCoder (LAPS)). This finding suggests that the LOGO domain is particularly well-suited to the token-to-token assumptions made by the IBM translation model from Wong et al.. It is also worth noting that only the DreamCoder and LILO conditions, which train a CNN-guided neural recognition model as part of enumerative search, have the ability to condition on the LOGO drawings. In particular, the conditions that rely exclusively on LLM-guided search must infer what to draw solely based on the task descriptions; an exceedingly difficult generalization task.

## B.4 EXPERIMENTS WITH HUMAN LANGUAGE DESCRIPTIONS

Each of our domains provides a default set of language task descriptions that were generated synchronously with the ground truth program(s) for each task. Following Wong et al. (2021), we use these synthetic language annotations for our primary experiments, as these descriptions correspond closely and systematically to the target programs. To test generalization to real-world applications, we also evaluated our methods on human language annotations sourced from Mechanical Turk. These were collected by Wong et al. (2021), with the exception of the REGEX domain, for which the annotations were sourced from the original (Andreas et al., 2018).

We ran experiments with a key subset of model conditions to compare performance on human vs. synthetic language. Fig. 13 and Tab. 5 summarize the results from these experiments. In general, synthesis performance with human language is upper-bounded by performance with synthetic language. This is expected, as the human language contains a wide range of lexical and syntactic variations. For instance, for an individual LOGO task involving drawing a snowflake, human annotations range from "3 sided snowflake with arms that are lines with a semi circle at the end" to "3 candy cane shapes with spaces in them," with one annotator simply stating, "boomerang." Compared to the more templated synthetic language, the broad variation present in the human annotations makes it more difficult to infer a mapping between the language and target programs.

Our experiments reveal that both search and library learning appear to be important to achieving robust performance on human language. DreamCoder achieves remarkably consistent performance between the two language types. In contrast, the LLM Solver baseline degrades markedly on CLEVR and LOGO with human descriptions. We see that adding search [LLM Solver (+ Search)] helps to mitigate this gap. Introducing the full library learning pipeline [LILO] further improves robustness to human language, while achieving better overall performance than DreamCoder.

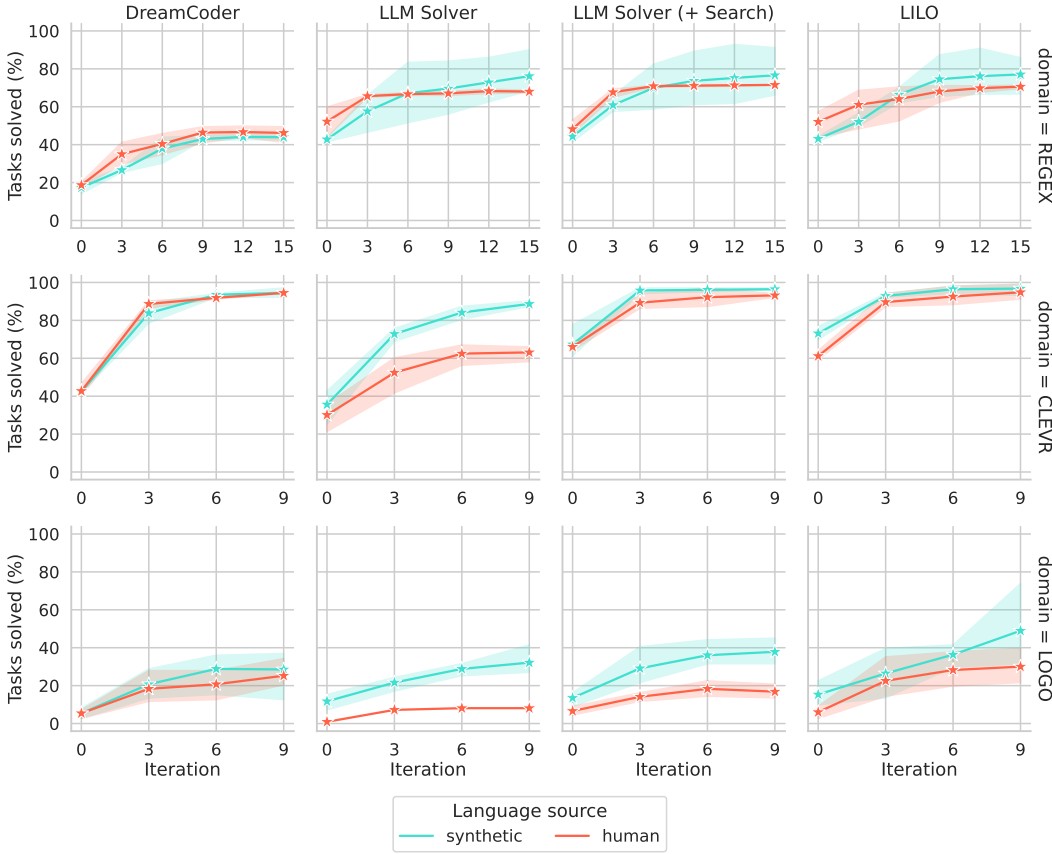

Figure 13: Learning curves illustrating performance of select models on human vs. synthetic language annotations.

| | SYNTHETIC LANGUAGE - TASKS SOLVED (%) | | | | | | | | |
| | REGEX | | | CLEVR | | | LOGO | | |
| MODEL | max | mean | std | max | mean | std | max | mean | std |
|---|---|---|---|---|---|---|---|---|---|
| DreamCoder | 45.60 | 43.93 | 1.53 | 97.09 | 94.50 | 2.44 | 36.94 | 28.53 | 13.79 |
| LLM Solver | 90.00 | 76.13 | 12.04 | 90.29 | 88.67 | 1.48 | 41.44 | 32.13 | 8.07 |
| LLM Solver (+ Search) | 91.20 | 76.60 | 13.02 | 97.09 | 96.44 | 0.56 | 45.05 | 37.84 | 6.80 |
| LILO | 93.20 | 77.07 | 14.14 | 99.03 | 96.76 | 3.12 | 73.87 | 48.95 | 22.15 |
| | HUMAN LANGUAGE - TASKS SOLVED (%) | | | | | | | | |
| DreamCoder | 49.40 | 46.20 | 4.39 | 95.15 | 94.50 | 0.56 | 34.23 | 25.23 | 7.85 |
| LLM Solver | 68.60 | 68.00 | 0.60 | 66.02 | 63.11 | 4.23 | 8.11 | 8.11 | — |
| LLM Solver (+ Search) | 71.60 | 71.53 | 0.12 | 94.17 | 93.20 | 0.97 | 20.72 | 16.82 | 3.64 |
| LILO | 71.40 | 70.60 | 0.92 | 99.03 | 94.82 | 3.92 | 39.64 | 30.03 | 9.07 |

Table 5: Solution rates of select models on human vs. synthetic language annotations.

## C  ADDITIONAL EXPERIMENTS AND ANALYSES

### C.1  COMPUTATIONAL EFFICIENCY ANALYSIS

Given that program search is the most computationally expensive component of synthesis, we would like to be able to quantify and compare the compute costs of LLM-based and enumerative search. However, performing an apples-to-apples comparison is non-trivial because the source of these costs is different between the two cases. As discussed in A.4, enumerative search requires a high degree of CPU parallelism, so the primary cost associated with running DreamCoder in our experiments is the on-demand CPU-hour cost of renting suitably large machines from AWS. In contrast, LLM search is GPU-intensive, and (in our implementation) is performed on external servers for which we do not have access to exact specifications or cost metrics. In practice, "LLM-as-a-service" models, such as OpenAI's API, charge a fixed price per text token, so the primary costs of LILO-style program search arise from the number of LLM queries, the length of the prompts, and the desired completion length.

In this section, we compare the computational efficiency of the two search approaches across three fronts. First, we consider *wall clock time*, which—in addition to being an informative metric in its own right—also allows us to compute a cost basis for enumerative search. Next, we consider *token usage*, which allows us to compute a cost basis for LLM search methods. These analysis culminate in a *dollar-to-dollar comparison* that, while dependent on pricing schemes of third-parties and the markets more generally, nevertheless offers the closest means of direct comparison.

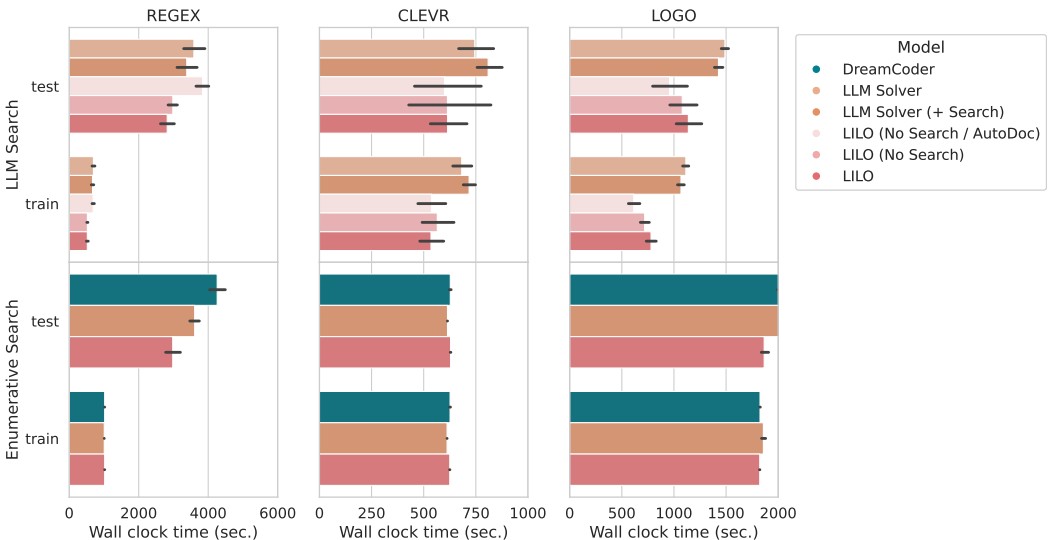

Figure 14: **Comparison of wall clock runtimes across search procedures and domains.** Each bar shows average runtime for a single iteration of train/test program search (error bars indicate 95% confidence intervals). Even with network latency from interfacing with OpenAI servers, LLM search (top row), typically requires less execution time than enumerative search (bottom row), which runs locally on a 96-CPU machine.

We start by analyzing observed (a.k.a. "wall clock") runtimes of our different models. Fig. 14 breaks these down by domain, where the x-axis corresponds to the average time to perform a single search iteration during training and test.[2] Overall, we observe that even with network latency from interfacing with OpenAI servers, a round of LLM search typically runs more quickly than an equivalent round of enumerative search. This difference is especially pronounced on LOGO, which requires longer search times (the enumeration budget for the DreamCoder baseline is set on a per-domain basis using the timeouts from Wong et al. (2021); see A.4 for more details). We do not observe

---

[2]Note that in Fig. 14, despite appearances, for a given model on a given domain, the per-task search times between train and test splits are approximately equal. Any apparent within-condition discrepancies between train and test are due to the fact that during training, we search on minibatches of 96 tasks, whereas during test, we search on the entire test set. Thus, for domains where the number of tasks is many multiples of the batch size (e.g., REGEX), there is a larger discrepancy betweeen train and test search times.

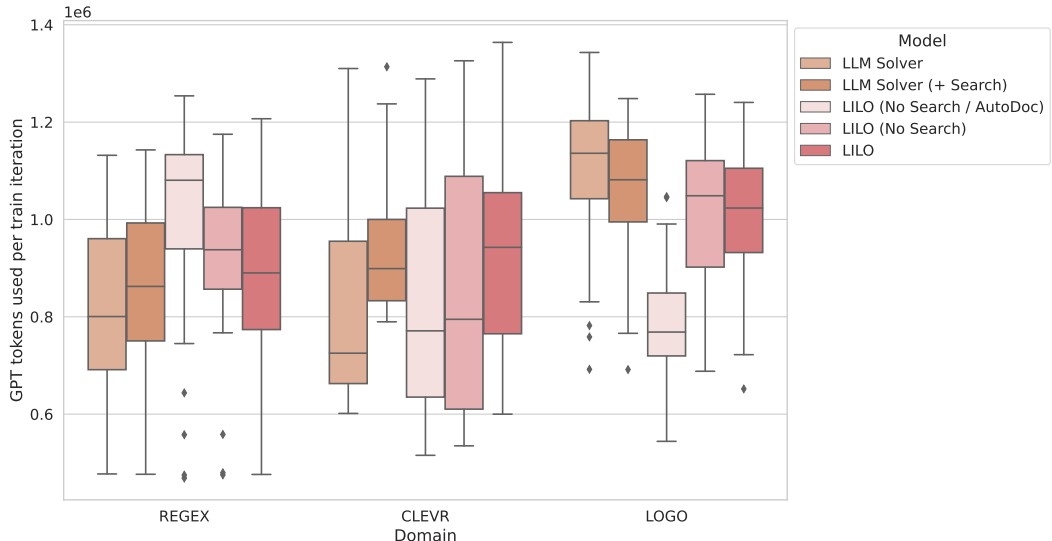

Figure 15: **GPT token usage per training iteration.** Token usage provides a useful metric for assessing the computational costs of LLM-based program search. A typical training iteration uses on the order of 0.8M-1.2M GPT tokens between the prompt and the completion. (Note the y-axis measures millions of tokens.) Boxes indicate quartiles of the distribution and whiskers extend to 1.5 inter-quartile ranges, with outliers shown as individual points.

major differences in runtimes within the different LLM Search conditions, though it is worth noting that the LILO and LLM Solver (+ Search) conditions require approximately 2x more total runtime than the other models because they perform both LLM-based and enumerative search on each iteration.

Next, we consider the token usage of the LLM solver conditions. Fig. 15 breaks these down by domain and model. A typical training iteration uses on the order of 0.8M-1.2M GPT tokens between the prompt and the completion. For completeness, all models are shown separately, but we do not note any clear trends in token usage by model; all models empirically use similar token counts. This may be because token usage is influenced by a complex interplay of several factors. Better-performing models will require fewer queries per task to discover a solution, so they should use fewer tokens. (In practice, however, we cap $n_{\mathrm{prompts\_per\_task}} = 4$, and all conditions must make at least one query per task, so the number of queries is bounded fairly tightly.) Models that use Stitch for compression (i.e., everything except the LLM Solver models) will also tend to benefit from shorter program description lengths per task. In particular, the LILO (– Search / AutoDoc) condition, which uses anonymous function names (e.g., fn_42), tends to use the fewest tokens per task. However, because we "pack the prompt" with as many examples as can fit, per-task description length does not directly influence token usage; though, as we discuss throughout, too much compression could affect token usage indirectly by obfuscating program semantics, therefore making the LLM solver require more queries to solve new tasks.

Finally, in the spirit of providing an apples-to-apples compute cost comparison, we combine our time cost and token cost analyses to estimate a dollar cost for each model per training iteration. For conditions that perform enumerative search, we compute CPU cost using the on-demand AWS EC2 instance price for a c5.24xlarge machine in us-east-2, currently priced at $4.08 / hr. Meanwhile, for conditions that involve LLM search (everything except DreamCoder), we compute LLM inference cost using OpenAI's current API pricing. As discussed in §3, our experiments took advantage of OpenAI's Codex model beta for researchers—in other words, they were effectively free. Accordingly, we estimate the cost of our queries using OpenAI's more recent gpt-3.5-turbo model, which is available to the public and priced at $0.002 per 1K tokens (at the time of writing). For the LLM solver cost analysis, we choose not to factor in the cost of running a "head node" to issue API queries, as this machine is an order of magnitude cheaper than the c5.24xlarge, has no specific spec requirements, and could be arbitrarily downscaled or even replaced with a laptop.

|  |  | REGEX | | CLEVR | | LOGO | |
| --- | --- | --- | --- | --- | --- | --- | --- |
|  |  | *mean* | *std* | *mean* | *std* | *mean* | *std* |
| LLM Search | LLM Solver | $1.65 | $0.35 | $1.66 | $0.44 | $2.19 | $0.32 |
|  | LLM Solver (+ Search) | $1.70 | $0.33 | $1.88 | $0.29 | $2.13 | $0.30 |
|  | LILO (✂ Search / AutoDoc) | $2.04 | $0.39 | $1.66 | $0.47 | $1.59 | $0.24 |
|  | LILO (✂ Search) | $1.86 | $0.30 | $1.70 | $0.52 | $2.03 | $0.31 |
|  | LILO | $1.77 | $0.38 | $1.87 | $0.42 | $2.01 | $0.30 |
| Enumerative Search | DreamCoder | $1.16 | $0.01 | $0.71 | $0.01 | $2.07 | $0.01 |
|  | LLM Solver (+ Search) | $1.14 | $0.00 | $0.69 | $0.00 | $2.11 | $0.06 |
|  | LILO | $1.16 | $0.00 | $0.71 | $0.00 | $2.07 | $0.00 |

Table 6: **Dollar cost comparison between LLM-based and enumerative search.** Each entry is the cost of running one training iteration of search, estimated based on measured wall-clock time (for enumerative search) or token usage (for LLM search). As a rough heuristic, we find that one iteration of LILO's LLM-amortized search scheme is approximately equivalent to an 1800-second enumerative search on 96 CPUs—or, about 48 CPU-hours—in terms of compute cost.

Tab. 6 summarizes the results of this analysis. Remarkably, despite the fact that LLM-based and enumerative searches use very different compute platforms with prices set by two different third-party companies, the dollar costs per training iteration come out to within the same order of magnitude— indeed, they are approximately comparable. In general, we find the tradeoff between LLM and enumerative search to be closely tied to the search time budget: domains with shorter enumeration timeouts (e.g., CLEVR) cost 2-2.5x less than LLM search, while domains with longer enumeration timeouts (e.g., LOGO) cost about the same. Therefore, as a rough heuristic, we can say that one iteration of LILO's LLM-amortized search scheme is approximately equivalent to an 1800-second enumerative search on 96 CPUs—or, about 48 CPU-hours—in terms of compute cost.

Of course, this cost analysis is heavily tied to market factors that are subject to change—in particular, the hardware, electricity, and logistics costs that AWS and OpenAI face in operating their compute platforms, as well as the profit margins that their pricing schemes bake in. Nevertheless, we find it noteworthy that it is currently possible to implement a search scheme like LILO—which requires thousands of LLM queries over millions of tokens per training iteration—while generally achieving better solution rates, faster wall clock runtimes, and comparable dollar costs to enumerative search. Moreover, we note that general-purpose cloud compute platforms like AWS have been available for many years; especially as Moore's Law is believed to be reaching its tail end (Theis & Wong, 2017), we are unlikely to see significant reductions in the cost of large-scale CPU compute. In constrast, the LLM-as-a-service model is a recent innovation; with increased scale, hardware optimizations, product maturation, and growing market competition, we are likely to see the costs of LLM inference decrease dramatically in the coming years. We are particularly excited about the growing diversity of open source LLM packages, which should make it possible to implement LILO in an even more cost efficient manner and with increased control over cost-performance tradeoffs.

### C.2    CAN LLMs PERFORM COMPRESSION?

In §4, we found that a pre-trained LLM is able to successfully perform the role of a program synthesizer, generating programs expressed in lambda calculus to solve novel tasks. Accordingly, it is natural to ask whether a LLM can also assume the role of a compression algorithm, generating useful and reusable abstractions to form the library. This question is central to the neurosymbolic claims of this work: if a LLM can be used for compression, then we could implement a version of LILO where all three modules in the framework are parametrized by LLMs.

Here, we set out to answer this question by benchmarking GPT-4's ability to generate useful abstractions when prompted with a set of programs. We began by constructing a procedurally-generated prompt format (Fig. 18) that combines prompting techniques used for LLM-guided synthesis and AutoDoc.[3] As with LLM-guided synthesis, context window limits require subsampling exemplars from the set of solved tasks. We explored two distinct subsampling methodologies:

1. **Random sampling** selects a fixed number of tasks without replacement. In our experiments, each prompt incorporated 30 program examples.
2. **Clustered sampling** divides the set of solved tasks into $k$ clusters of semantically-related tasks and attempts to generate at least one abstraction for each cluster. We first generated embeddings for all task descriptions using the `text-embedding-ada-002` from OpenAI. Next, we ran k-means clustering to group all solved tasks into a fixed number of clusters based on these embeddings. In our experiments, we set $k = \max(10, \lfloor \frac{N}{10} \rfloor)$, where $N$ is the total number of solved tasks at a particular iteration, to ensure each cluster contained at least 10 examples. Finally, for each cluster, we subsampled a maximum of 30 examples to use in the prompt.

We evaluated our **LLM Compressor** against Stitch in a controlled compression experiment using both sampling methods on all three domains. At each iteration, we provided both compression procedures with an identical frozen set of programs solved by the LLM Solver in our online synthesis experiments from §4. Next, we rewrote the set of solved training programs using the generated abstractions to compute the **compression ratio**: a scalar value in $[1.0, \infty)$ that indicates the overall reduction in corpus length afforded by a particular library.

Fig. 16 shows the primary results from these experiments. Across all three domains, Stitch achieved compression ratios in the 1.5-3.5x range, with performance generally improving as more solved

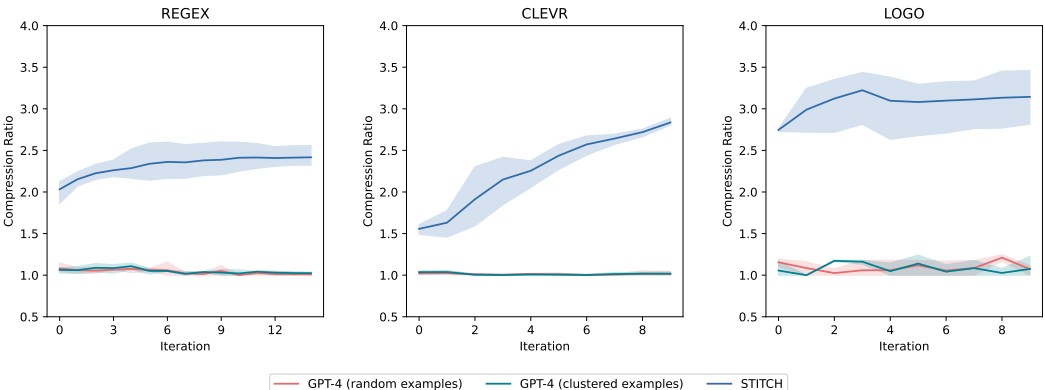

Figure 16: **Comparison of compression ratios achieved by GPT-4 vs. Stitch during online synthesis.** Stitch produces abstractions that achieve 1.5-3.5x compression of the set of solved tasks, with performance generally improving over the course of learning. In contrast, GPT-4 struggles to generate abstractions that achieve meaningful compressivity.

---

[3]As is generally the case when attempting to induce LLMs to perform a novel task, this exercise required exploring a theoretically infinite space of possible prompts. Here, our standard is "best effort prompt engineering": we set out to demonstrate a positive result and spent a significant amount of time iterating on the prompt format, drawing on techniques developed for other components of LILO. Compared to the LLM Solver prompt (Fig. 6), this task required significantly more prompt engineering effort and yielded only negative results.

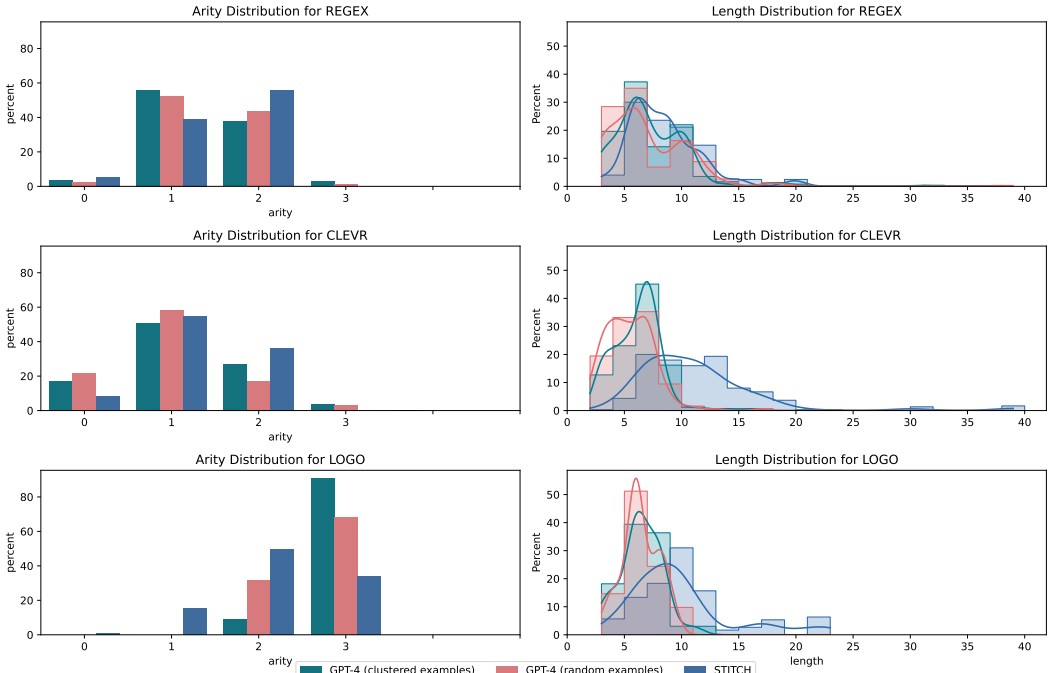

Figure 17: **Formal characteristics of abstractions produced by GPT-4 vs. Stitch.** Left: Comparison of abstraction arity (number of arguments). Right: Comparison of abstraction length (in terms of number of program primitives). Stitch is able to produce *longer* abstractions that encapsulate more reusable logic, while GPT-4 tends to generate trivial abstractions.

programs become available. In contrast, the LLM Compressor struggled to produce abstractions that achieved meaningful compressivity, hovering around the 1.0x baseline. Moreover—and unlike the LLM Solver—the LLM Compressor had difficulty producing syntactically well-formed expressions, with only 19.01% (998 / 5250) of the generations across the three DSLs representing valid expressions (Tab. 7).

We also evaluated two formal characteristics of the generated abstractions: length and arity (Fig. 17). *Length* is the number of base DSL primitives used in the abstraction and *arity* is the number of arguments. We compare the distribution of length and arity among all abstractions generated using Stitch and the LLM Compressor. While both methods produce similar arity distributions, the abstractions produced by Stitch tend to be longer on average (Fig. 17, right). This metric reflects the fact that Stitch

|  | Valid | Generated | Valid % |
|---|---|---|---|
| REGEX | 561 | 2250 | 24.93% |
| CLEVR | 363 | 1500 | 24.20% |
| LOGO | 74 | 1500 | 4.93% |
| **Overall** | **998** | **5250** | **19.01%** |

Table 7: **Validity of abstractions generated by LLM Compressor.**

produces abstractions that meaningfully capture shared logic, while GPT-4 tended to produce trivial abstractions. Overall, these results led us to decide not to conduct further and more costly online synthesis experiments with the LLM Compressor, as these abstractions appeared unlikely to aid in synthesis.

While claims of the form "GPT-X cannot perform compression" is unfalsifiable (see Footnote 3), these results nevertheless highlight the fact that *current LLMs are more readily-equipped to perform synthesis than compression*. Intuitively, the complexity of generating a valid abstraction is significantly higher than the equivalent generation task for programs, as it requires reasoning formally about how the proposed abstraction will interact with the entire corpus of programs. These results reinforce the motivation of LILO as a neurosymbolic framework that uses LLMs for synthesis while delegating refactoring to a symbolic module that has been engineered to handle this more computationally difficult task.

**Domain-general example** **A**

*You are about to undertake a task focused on abstraction learning. The objective is to generate reusable functions derived from existing ones. These new functions will be utilized to solve specific tasks. Here is an example for producing a reusable function for a simple numeric domain.*

*Input:*
```
(lambda (+ 3 (* (+ 2 4) 2)))
(lambda (map (lambda (+ 3 (* 4 (+ 3 $0)))) $1))
(lambda (* 2 (+ 3 (* $0 (+ 2 1)))))
```

*Observation: All the input programs add 3 to the result of a multiplication.*

*Function expression:* `(lambda (lambda (+ 3 (* $0 $1))))`

*Now, let's rewrite the input to use this function expression:*
```
(lambda (lambda (lambda (+ 3 (* $0 $1))) (+ 2 4) 2)
...
```
```
{
    "readable_name": "multiply_and_add_three",
    "function_expression": "(lambda (lambda (+ 3 (* $0 $1))))",
    "description": "Multiplies the two arguments and adds three to the result."
}
```

**DSL functions** **B**

*Here, we are working in a domain-specific language that includes the following library functions:*
```
regex_not :: tsubstr -> tsubstr
regex_or :: tsubstr -> tsubstr -> tsubstr
regex_concat :: tsubstr -> tsubstr -> tsubstr
...
```

**Example programs** **C**

*Consider some examples of solved tasks:*
```
(lambda (regex_flatten (regex_map (lambda (regex_if (regex_match 't' $0) (regex_concat 'a'
'x') $0)) (regex_split empty_string $0))))
(lambda (regex_flatten (regex_cons 'n' (regex_cons 'a' (regex_cdr (regex_split '.' $0))))))
(lambda (regex_flatten (regex_map (lambda (regex_if (regex_match 'p' $0) (regex_concat 'f'
'w') $0)) (regex_split '.' $0))))
<additional examples based on sampling method>
```

**Task instruct-ions** **D**

*Please write a compact, reusable function based on the programs above. Find patterns in the programs that can be reused across multiple tasks.*

*- Only propose one function.*
*- Only use the functions available to you in the library provided.*
*- Make sure the abstraction has balanced parentheses.*
*...*
*- Your response should use the following JSON format.*
```
{
    "readable_name": TODO,
    "function_expression": TODO,
    "description": TODO
}
```

Figure 18: **Anatomy of a LLM Compressor prompt.** The prompt consists of (A) a header demonstrating the abstraction task for a general numeric domain, (B) a DSL description enumerating the available library functions, (C) a set of example programs sampled using either random or cluster-based sampling (described above), and (D) a footer specifying constraints and the desired output format. The JSON schema for the LLM Compressor output is similar to the AutoDoc prompt (A.2) with the addition of a `function_expression` field.

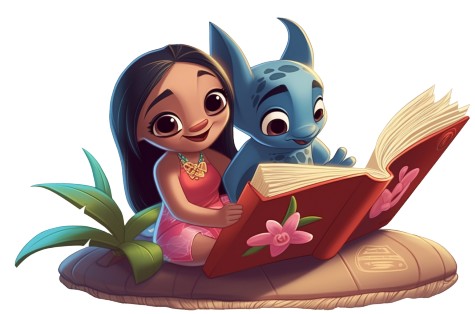

*Some of the graphical assets that appear in this work were generated by the authors via Midjourney, a third-party service provider. Subject to the Midjourney Terms of Service for Paid Users, all assets created with the services are owned by the authors to the extent possible under current law. The authors acknowledge and respect the copyrights and trademarks held by the Walt Disney Company. Any likeness to characters or properties trademarked by Disney is considered Fair Use under US Transformative Use laws, which provide broad protections for commentary, criticism, parody, satire, education, research, and other forms of creative expression.*

