# OpenReview forum: "LILO: Learning Interpretable Libraries by Compressing and Documenting Code"
_ICLR.cc/2024/Conference — ICLR 2024 poster_

### Official Review · Reviewer_dfJc · 2023-10-19

**Soundness:** 3 good
**Presentation:** 4 excellent
**Contribution:** 2 fair
**Rating:** 6
**Confidence:** 4

**Summary:**

The paper proposes a neuro-symbolic framework for learning "code libraries" for lambda-calculus code.
The idea is to iteratively synthesize code using an LLM, compress it using an existing approach ("Stitch") and then document it using an LLM.
By repeating this approach multiple times, the overall framework learns repetitive "libraries" that can be further useful in the next iteration.

The overall framework, called Lilo, is evaluated on three tasks: CLEVR (visual scene reasoning), REGEX (string editing with regular expressions), and LOGO (graphics composition in the 2D Logo graphics language), and shows improvements over a vanilla LLM and over DreamCoder (Ellis et al., 2021).

**Strengths:**

1. The idea is interesting, combining both LLMs and traditional program synthesize.
2. It is refreshing to see tasks and approaches in which using vanilla LLMs still works much worse than a clever approach.

**Weaknesses:**

1. The tasks are a quite contrived, and it feels like the problem definition is tailored exactly to this kind of solution. All the tasks are synthetic and based on lambda calculus. What practical problems can the proposed approach be helpful for? Who is expected to use the proposed approach?
2. Evaluation - I am not sure that it is fair to compare that proposed approach to a *prompted* LLM as a baseline when this baseline is prompted with the lambda-calculus language, which it probably was not trained on. Since the task is defined in a domain-specific-language (DSL), a prompted LLM is expected to fail. Can the task be "translated" to Python? If the LLM would be allowed to use Python, that would be a more fair comparison. Alternatively, a more fair baseline could be fine-tuning an LLM on the language, rather than just prompting, since the proposed approach also does kind-of training within its compression step.
3. Novelty - I am not familiar with the directly related work, and I am thus not sure about the novelty: the paper claims that the framework consists of three modules:
    1. A synthesis module - which is a prompted LLM
    2. A compression module ("Stitch"), which is adopted as is from [Bowers et al., 2023](https://mlb2251.github.io/stitch_jul11.pdf).
    3. An "auto-documentation module" which uses a prompted LLM to document the resulting code pieces, for better interpretability and improvement of the next iteration.

   It seems like steps (1) and (3) are standard code generation using LLM prompting, and the heavy lifting is done by the 2nd step. Since the 2nd step is identical to Bowers et al. (2023), I am not sure how novel is this paper compared to Bowers et al. (2023).

**Questions:**

### Questions

1. What practical problems can the proposed approach be helpful for? Who is expected to use the proposed approach?
2. Can the task be "translated" to Python, and then tested on an LLM?
3. Can an LLM such as LLaMA be finetuned on the training examples, to better understand their language? Prompting an LLM with a language it was not trained on is a bit unfair.

### Comments
1. Regarding the AutoDoc module - more than *having* an auto-documentation module, the interesting part to me is that this generated documentation improve the next iteration, when fed to the LLM. That is, the self-improvement part (the fact that the LLM generated doc improves the LLM itself in the next iteration) is the interesting from an ML perspective, to me, while the paper mostly emphasizes the "existence" of this module to generate documentation, which is not significantly novel or interesting on its own.

### Summary
The main weaknesses of the paper are its lack of practicality, the tasks are quite synthetic, and I believe that an LLM-based baseline can perform better than presented, if given fair chances.
Nonetheless, the paper proposes a simple approach (simpler than DreamCoder, which is good), it is well written, easy to follow, and the ideas are conceptually pleasing, and I thus vote for acceptance.

---

> ### Author Response · Authors · 2023-11-16
> **Thank you for your positive review!**
>
> We are glad that you found the ideas in the paper to be “interesting” and “conceptually pleasing” and that you found the overall paper to be “refreshing” and “well-written.” Thank you for your vote for acceptance.
>
> Please allow us to address the points that you raised below:
>
> **The tasks are a quite contrived, and it feels like the problem definition is tailored exactly to this kind of solution. All the tasks are synthetic and based on lambda calculus.**
>
> It is true that some program synthesis tasks have the synthetic quality that you are describing (as Reviewer NLqL points out, this is “a common limitation of program synthesis models/systems”). However, we don’t feel it is accurate to say that these tasks have somehow been tailored to our solution—**we’ve selected a canonical set of benchmarks that have been studied extensively in prior program synthesis work** [1, 2]. This affords the ability compare performance against prior results (see Appendix B.3) to a degree that we would not have been able to had we constructed an original benchmark. Evaluating on a novel benchmark would also have been more likely to raise concerns about “tailoring” of the kind described here.
>
> **What practical problems can the proposed approach be helpful for? Who is expected to use the proposed approach?**
>
> First, we want to start by addressing a possible misperception that systems based on DSLs of the kind we study here lack real-world application. Ideas from this line of inductive program synthesis research have resulted in many high-profile applications over the years, including:
>
> - **Tabular autocompletion** [5], which has powered Microsoft’s billion dollar Excel product line since 2013 and continues to do so a decade later [6]. The REGEX domain in our work is a restricted version of the DSL used for systems like FlashFill.
> - **Scientific data analysis**, which has been applied to analyze diverse phenomena ranging from large scale video datasets of social behavior in laboratory mice [7, 8] to protein binding and activity prediction for RNA splicing [9]. Our CLEVR domain, which requires reasoning about spatial relationships between objects in scenes, presents many of the same challenges needed for analyzing this kind of graph-structured data.
> - **Long-horizon robotics tasks** of the kind that are performed by industrial/warehouse robots [10]. Our LOGO domain, which requires synthesizing programs to instruct a “turtle” how to move on a 2D surface by specifying angles and velocities, can be viewed as a version of the kind of planning problems involved in robotics domains.
>
> These recent advances also a key part of the motivation for the neurosymbolic & hybrid AI systems track at ICLR this year. Situated in this context, the ideas and methods we introduce in LILO should have broad appeal, not just for this audience, but for the ML community in general. In particular, we believe that LILO’s general design pattern around synthesis, refactoring, and documentation has relevance to practical software engineering tools like Copilot that are used by millions of programmers every day.
>
> Practically speaking, our code implementation (withheld for anonymity) has already garnered attention from the GitHub community and students at multiple universities are presently using LILO as the basis for research projects. Concretely, we put significant effort into making the LILO codebase beginner-friendly through documentation and by providing (for the first time in this line of work) a Dockerized development environment. Lastly, we strongly resonate with your assessment that LILO is **“simpler than DreamCoder, which is good.”**

---

> > ### Author Response · Authors · 2023-11-16
> >
> > **Can a LLM (e.g., LLaMA) be finetuned on the DSL?**
> >
> > We want to emphasize that in our “lifelong learning” evaluation setting, which follows from DreamCoder, **the number of program examples available at the start of learning is very small**—about 10-30 examples, which is likely not enough for finetuning a LLM. Moreover, the goal of this data scarcity is to test models’ ability to perform humanlike generalization from a small handful of examples. Under this view, the pretrained LLM in LILO plays the role of a broad prior that captures some of the common knowledge that a human programmer might bring to a novel problem domain. While it is possible to collect a larger set of annotated examples to finetune a LLM, we believe this is a different kind of evaluation setting that is both less novel and also diverges in motivation from the goals of this line of work.
> >
> > **Prompting a LLM with a programming language it was not trained on is a bit unfair.**
> >
> > We use the same few-shot prompting setup for LLM-guided synthesis across all models, including LILO; therefore, any code comprehension issues that affect the LLM baselines will also affect LILO in the same ways. Conversely, if finetuning a LLM on one of our DSLs were to raise the LLM Solver baseline, it would also necessarily improve LILO’s performance. For this reason, we don’t agree with the assessment that there is something “unfair” about the design decision to use few-shot prompting instead of finetuning.
> >
> > We also want to address the misperception that these DSLs are inherently problematic for a 175B parameter LLM like `code-davinci-002` as this has not been our experience. The DSLs in this work all use LISP syntax, which is prevalent across many modern languages (Scheme / Racket / Clojure / etc.) and are therefore well-represented in code pre-training datasets like GitHub. The main aspect the LLM will be unfamiliar with is the semantics of the DSL, which is precisely the point — our goal is to build a model that can generalize to novel domains based on only a handful of examples and without expensive finetuning.
> >
> > **Can the tasks be translated into Python?**
> >
> > Yes—and this is something that we considered early in the project. In practice, however, we found that syntax errors with Codex were rare (e.g., virtually no issues with parenthesis matching). Given that the LLM Solver already showed strong performance on the online synthesis experiments when working in the original syntax, we opted not to introduce additional system complexity of translating the programs into a Python-like syntax. Irrespective of the syntax, translation would not address the fact that the main difficulty for LLM-guided synthesis arises from the novel semantics of the domains and the novel abstractions introduced by compression. Accordingly, our emphasis in this work is on methods to address issues with semantic comprehension (i.e., via AutoDoc) and not syntactic comprehension.

---

> > > ### Author Response · Authors · 2023-11-16
> > >
> > > # Novelty and contributions
> > >
> > > As a framework, LILO’s novelty lies just not just in the individual components of the system, but in the connections between them. As Reviewers oPbs notes,
> > >
> > > > “In terms of contribution, the paper situates itself as a complete system that offers an improvement over DreamCoder across a varied collection of tasks; I think the evidence the paper presents on this point is compelling.”
> > >
> > > We would like to highlight several novel contributions introduced in LILO through integration of ideas from multiple research literatures:
> > >
> > > - **Application of LLMs to program synthesis in traditional DSLs:** Much of the recent work on LLMs code generation focuses on well-represented languages (e.g., Python synthesis on the HumanEval benchmark). This work has been largely disconnected from decades of prior work in inductive program synthesis. LILO represents an important step in closing this gap by evaluating LLM-guided synthesis directly on prior benchmarks using lambda calculus DSLs. As Reviewer oPbs observes, “in some ways the most surprising results from the paper are that, even without library learning, LLMs can solve these domain-specific tasks much better than prior works.” This is also a reason why we opted not to translate our tasks into Python—part of the express goal of this work is to evaluate the feasibility of synthesizing programs in these traditional DSLs using LLMs.
> > > - **Integration of LLMs with a symbolic compression module:** The idea of integration between LLMs and symbolic tools has gained a lot of popularity in the LLM community recently; e.g., through works like Toolformer [3]. Within the “LLMs for programming space,” existing work has focused on integrations of LLMs with interpreters, databases, symbolic logics, and other kinds of specialized APIs. Here, we emphasize the opportunity to integrate with an external symbolic compression module to perform *refactoring.* To our knowledge, this integration of a LLM with a symbolic code compression tool is the first of its kind and offers a new design pattern for practical software engineering tools like Copilot.
> > > - **Application of Stitch to the online program synthesis setting:** Related to the above, while the algorithmic contributions of Stitch are prior work, LILO represents a novel *application* of Stitch—not just to new domains, but to a new, more ecologically-relevant synthesis setting. The experiments in [4] were all performed on static corpora of pre-annotated programs, with the evaluations focusing on benchmarking the compressivity and performance of Stitch relative to DreamCoder’s compression algorithm. In this work, we perform the first experiments using Stitch as part of an online program synthesis setting, demonstrating its ability to function as part of a broader library learning loop. We also apply Stitch to three new domains that were not considered in the original work.
> > > - **Continual self-improvement via auto-documentation:** Our results demonstrate that LILO is able to bootstrap LLM synthesis performance via auto-documentation. We are glad that you found this result “interesting from an ML perspective” and agree that this is one of the most exciting selling points of this work. We certainly intended to emphasize the self-improvement aspect of AutoDoc throughout the paper; e.g.,
> > >     - “In addition to improving human readability, we find that AutoDoc boosts performance by helping LILO’s synthesizer to interpret and deploy learned abstractions.” (p. 1, abstract)
> > >     - “yielding better interpretability and improving downstream LLM-guided search” (p. 1)
> > >     - “ Key to this neurosymbolic integration is our AutoDoc module, which not only improves interpretability, but also helps the LLM synthesizer use the library more effectively” (p. 2)
> > >     - “Unlike traditional program synthesis methods, language models draw inferences about the semantics of programs from their lexical content… we explore how AutoDoc benefits downstream synthesis performance” (p. 5)
> > >     - “Auto-documentation unlocks effective contextual usage of abstractions.” (p. 8)
> > >
> > > All of the above are intended to emphasize the functional benefits of the AutoDoc module beyond simply pointing to its existence in the architecture. Nevertheless, we certainly want to make sure this point is coming through. We will take your comment under advisement and look for ways to further emphasize this novel aspect in our revised submission.

---

> > > > ### Author Response · Authors · 2023-11-16
> > > >
> > > > [1] Wong, C., Ellis, K. M., Tenenbaum, J., & Andreas, J. (2021, July). Leveraging language to learn program abstractions and search heuristics. In *International Conference on Machine Learning* (pp. 11193-11204). PMLR.
> > > >
> > > > [2] Andreas, J., Klein, D., & Levine, S. (2018, June). Learning with Latent Language. In *Proceedings of the 2018 Conference of the North American Chapter of the Association for Computational Linguistics: Human Language Technologies, Volume 1 (Long Papers)* (pp. 2166-2179).
> > > >
> > > > [3] Schick, T., Dwivedi-Yu, J., Dessì, R., Raileanu, R., Lomeli, M., Zettlemoyer, L., ... & Scialom, T. (2023). Toolformer: Language models can teach themselves to use tools. *arXiv preprint arXiv:2302.04761*.
> > > >
> > > > [4] Bowers, M., Olausson, T. X., Wong, L., Grand, G., Tenenbaum, J. B., Ellis, K., & Solar-Lezama, A. (2023). Top-down synthesis for library learning. *Proceedings of the ACM on Programming Languages*, *7*(POPL), 1182-1213.
> > > >
> > > > [5] Gulwani, S. (2011). Automating string processing in spreadsheets using input-output examples. *ACM Sigplan Notices*, *46*(1), 317-330.
> > > >
> > > > [6] Cambronero, J., Gulwani, S., Le, V., Perelman, D., Radhakrishna, A., Simon, C., & Tiwari, A. (2023). FlashFill++: Scaling programming by example by cutting to the chase. *Proceedings of the ACM on Programming Languages*, *7*(POPL), 952-981.
> > > >
> > > > [7] Shah, A., Zhan, E., Sun, J., Verma, A., Yue, Y., & Chaudhuri, S. (2020). Learning differentiable programs with admissible neural heuristics. *Advances in neural information processing systems*, *33*, 4940-4952.
> > > >
> > > > [8] Segalin, Cristina, Jalani Williams, Tomomi Karigo, May Hui, Moriel Zelikowsky, Jennifer J. Sun, Pietro Perona, David J. Anderson, and Ann Kennedy. "The Mouse Action Recognition System (MARS) software pipeline for automated analysis of social behaviors in mice." *Elife* 10 (2021): e63720.
> > > >
> > > > [9] Gupta, K., Yang, C., McCue, K., Bastani, O., Sharp, P., Burge, C., & Lezama, A. S. (2023). Improved modeling of RNA-binding protein motifs in an interpretable neural model of RNA splicing. *bioRxiv*, 2023-08.
> > > >
> > > > [10] Patton, N., Rahmani, K., Missula, M., Biswas, J., & Dillig, I. (2023). Program Synthesis for Robot Learning from Demonstrations. *arXiv preprint arXiv:2305.03129*.

---

> > > > > ### Comment · Reviewer_dfJc · 2023-11-21
> > > > > **Response to authors**
> > > > >
> > > > > Thank you for your response.
> > > > >
> > > > > I am still concerned about the applicability of these tasks. Excel's FlashFill has been used by the program synthesis community to justify these kinds of tasks over a decade, while machine learning research had gone through groundbreaking breakthroughs in the meantime.
> > > > > In other words, I think that these tasks are defined in such a specific and unnatural way to allow new approaches to make progress, but they have lost connection to the real world long ago.
> > > > >
> > > > > However I agree that these benchmarks are canonical and have been studied extensively in prior program synthesis work, and the authors should not be penalized for it.
> > > > >
> > > > > Regarding the potential translation to Python:
> > > > > >Can the tasks be translated into Python?
> > > > >
> > > > > >Yes—and this is something that we considered early in the project. In practice, however, we found that syntax errors with Codex were rare (e.g., virtually no issues with parenthesis matching). Given that the LLM Solver already showed strong performance on the online synthesis experiments when working in the original syntax, we opted not to introduce additional system complexity of translating the programs into a Python-like syntax. Irrespective of the syntax, translation would not address the fact that the main difficulty for LLM-guided synthesis arises from the novel semantics of the domains and the novel abstractions introduced by compression. Accordingly, our emphasis in this work is on methods to address issues with semantic comprehension (i.e., via AutoDoc) and not syntactic comprehension.
> > > > >
> > > > > I don't understand why the fact that syntax errors with Codex are rare matters? I would argue that even if Codex can generate syntactically valid DSL, it would still perform much better semantically if the input was Python.
> > > > > The benefit of models such as Codex in processing Python-translated inputs goes far beyond syntax only.
> > > > >
> > > > > I am left with the conclusion that in the way these benchmarks are currently defined - the approach is useful, and thus the paper should be accepted.
> > > > > However, if anyone really wanted to solve the corresponding real-world problems, they could just translate the problems to Python, and the marginal benefit of approaches such as Lilo and DreamCoder would be minimal.

---

### Official Review · Reviewer_NLqL · 2023-10-23

**Soundness:** 4 excellent
**Presentation:** 3 good
**Contribution:** 4 excellent
**Rating:** 8
**Confidence:** 4

**Summary:**

The paper presented a dual strategy system utilizing LLM and symbolic systems for program synthesis. Unlike conventional programming synthesis models, which implicitly learn the common program fragment (library), the proposed LILO explicitly generates a library with human-readable function names. The code generator is informed with the library and is able to generate potentially more concise program.

The experimental results show its improvement against the straightforward LLM + search approach and the LLM free baseline, DreamCoder.

**Strengths:**

The work demonstrated a successful combination of LLM and symbolic tools to efficiently generate a library and set of programs given a DSL and program synthesis tasks.

**Weaknesses:**

As the author also mentioned, the pipeline needs an additional self-verification step for the auto-documentation. The validity and conciseness of the language-guided program synthesis is enforced by input-output pairs and the stitch compression, while the library auto-doc has no verification or feedback loop to update the LLM or other trainable part of the pipeline.

The application of the method is limited in functional programming with relatively simple DSL. Though it's almost a common limitation of program synthesis models/systems.

**Questions:**

1. Could the author elaborate on algorithm 1 and Dual-system program search? The relationship between the enumerate search and LLM-guided search is quite unclear. Are they parallel? cascaded? or combined in another manner?

2. What does PCFG stand for? Could the author describe the input-output, architecture, and training of this multi-layer perceptron? Is this model pre-trained? DSL-specific?

3. In the Table 1, only in LOGO, the LILO method significantly outperforms the LLM solver + search. Could the author explain the performance difference among the different DSLs? If the reason is that LLM is not pre-trained on only LOGO, it could be an interesting and promising observation about the generalization capability of both LLM and LILO.

---

> ### Author Response · Authors · 2023-11-16
> **Thank you for your positive review!**
>
> We are glad that you found this work to demonstrate a “successful combination of LLM and symbolic tools” and appreciate the strong rating for acceptance.
>
> Thank you for the valuable feedback! **Based on your questions, we’ve updated several areas of Section 3 to clarify details about the search module.** We hope these changes—and the clarifications below—will address any issues you encountered in the “Presentation” category.
>
> # Response to questions
>
> **In the dual-system program search module, what is the relationship between the enumerative search and LLM-guided search?**
>
> Good question! It’s a straightforward serial implementation: first, we run LLM-guided search as a fast “System 1” first pass (Alg. 1, lines 5-6) and then run enumerative search as a slow “System 2” process to try to solve tasks that were not solved by the LLM. We agree that this is not as clear as it could be; **we will update Sec. 3 of our submission to emphasize that LLM-guided search and enumerative search are separate procedures.**
>
> Various approaches have been proposed for combining neural and symbolic search methods (e.g., [1] for a review), but most of this work is from the pre-LLM era. Smarter ways of integrating LLMs and enumeration for program search is something we’re currently exploring — of particular interest are methods for grammar-constrained generation with LLMs (e.g., [2]) — but this is probably beyond the scope of the current paper.
>
> Lastly, it’s worth noting that DreamCoder’s enumerative search, which we use in LILO, trains a neural network to predict PCFG weights, so “enumeration” is itself an instance of neurally-guided symbolic search. We provide more detail on this below, in response to your questions.
>
> **What does PCFG stand for?**
>
> A probabilistic context free grammar (PCFG) is a generalization of a context-free grammar that assigns a real-valued weight to each production rule. **We define this term towards the start of the Preliminaries section on p. 3**: “the prior is defined under a probabilistic context free grammar (PCFG; Johnson 1998)...”
>
> More formally, a PCFG can be defined as a quintuple $G = (M, T, R, S, P)$ where $M$ is a set of non-terminal symbols, $T$ is a set of terminal symbols, $R$ is a set of production rules, $S$ is the start symbol, and $P$ is a function that assigns a probability to each rule in $R$. This probabilistic element allows for the generation of a diverse range of structures. Moreover, as we describe below, training a task-conditioned neural network to infer $P$ is a natural way to guide search in the PCFG.
>
> **Could the author describe the input-output, architecture, and training of this multi-layer perceptron? Is this model pre-trained? DSL-specific?**
>
> Happy to provide more clarification! Here, we are talking about a piece of DreamCoder’s enumerative search algorithm that trains a **“recognition network”** (i.e., an RNN encoder + a MLP) to guide enumerative search by inferring $P$: the probabilities on production rules in the PCFG, as defined above.
>
> ***All of this machinery is part of prior work (i.e., DreamCoder)*** and uses off-the-shelf Pytorch components, so we don’t allocate much space in the main paper body to describe these details. Nevertheless, these are natural questions for a reader interested in implementation, so we provide an overview of the architecture below. **We refer to Appendix I in the [DreamCoder supplement](https://dl.acm.org/doi/10.1145/3453483.3454080) for further details details and will add a pointer to this appendix in the paper.**
>
> - **Inputs and outputs:** The recognition network takes as input a set of input/output examples for a specific task and produces an $|\mathcal{L}| \times |\mathcal{L}|$ tensor with values in $[0, 1]$ specifying transition probabilities for every production in the PCFG.
> - **Architecture/training:** The input/output examples are encoded using an RNN (specifically, a bidirectional GRU) for text-based domains and a CNN for image-based domains. The actual MLP module is very small (only two layers, with a hidden size of 64) so the whole architecture can be learned efficiently in-the-loop. During each learning iteration, the recognition network is trained for a maximum of 10K gradient steps.
> - **Pre-training:** As implemented, the model doesn’t use any pre-trained components. Following prior work with these datasets, we assume ground truth access to the CLEVR scene graphs, which allows us to use a text-based encoder for that domain. If we wanted to train on the original images, one could easily swap in a pre-trained CNN, or even a SOTA segmentation model, for the encoder.
> - **DSL-specificity:** Overall, the recognition network architecture is fairly general—the main domain-specific pieces are the choice of encoder and the dimensionality of the output, which depends on $|\mathcal{L}|$.

---

> > ### Author Response · Authors · 2023-11-16
> >
> > **Only in LOGO does LILO significantly outperform LLM solver + Search. Could the author explain the performance difference among the different DSLs?**
> >
> > First, to clarify, this is only the case for online synthesis—we note that **none of the results from offline synthesis (Table 1, bottom half) fall within $1\sigma$ of the best (bolded) result**. These include the two new baselines added during the response period. We also note that LILO’s $L_f$ consistently finds more solutions *faster* than the other libraries evaluated in offline synthesis.
> >
> > To address your question about LOGO: we find that LOGO is intrinsically more difficult than the other two DSLs across the board, for all models. We attribute this to the fact that many LOGO tasks require discovering domain-specific program structures (e.g., how to draw a “snowflake” or “staircase”; see Fig. 5) that require mastery of concepts like radial symmetry. Anecdotally, the authors also find LOGO to be the hardest DSL to write programs in!
> >
> > # Response to weaknesses
> >
> > **As the author also mentioned, the pipeline needs an additional self-verification step for the auto-documentation.**
> >
> > We generally agree, though we feel that “needs” is a bit of a strong characterization given that we show AutoDoc already helps with performance and is usually correct. However, this is a natural area for follow-up work. While this is not something that we expect to be able to implement during the review response period, it is easy to envision how such a “self-verificaiton” module might be architected following standard LLM prompting patterns.
> >
> > Additionally, we’ve started working on a small experiment comparing AutoDoc with expert-generated documentation in response to comments from Reviewer oPbs. This may be of interest as it touches on the same question you’ve raised here around whether stronger verification for documentation can improve performance.
> >
> > **The application of the method is limited in functional programming with relatively simple DSL.**
> >
> > As noted, these are common limitations of program synthesis research. There are good reasons why a lot of work from that community has historically focused on these settings. Functional languages have many properties that make them more amenable to research. For instance, the absence of side-effects are one language feature that make it possible to implement a program compression algorithm like Stitch. Currently, there is no equivalent in imperative languages like Python, though as we note in the last paragraph of Section 5, we view this as a promising and tractable area for future work in PL. Given an appropriate compression algorithm implementation, it would be straightforward to implement LILO for Python!
> >
> > [1] Chaudhuri, S., Ellis, K., Polozov, O., Singh, R., Solar-Lezama, A., & Yue, Y. (2021). Neurosymbolic programming. *Foundations and Trends® in Programming Languages*, *7*(3), 158-243.
> >
> > [2] Willard, B. T., & Louf, R. (2023). *Efficient Guided Generation for Large Language Models* (arXiv:2307.09702). arXiv. http://arxiv.org/abs/2307.09702

---

> > > ### Comment · Reviewer_NLqL · 2023-11-21
> > > **Thanks for the response**
> > >
> > > I acknowledge that I have read the authors' responses and other reviewer's discussion with the authors.
> > >
> > > Overall I believe this work is above the acceptance threshold of ICLR, and also above the quality of other papers I'm assigned in this venue, despite the domain specific limitation. Given the computation cost and tremendous pre-training/finetuning/prompting options in LLM, it's kind of hard to make really "fair" comparison now, so my judgement is more based on the completeness and robustness of the entire pipeline/system. From this aspect, the only concern I had was the auto-documentation, and I agree with the authors that it should be beneficial to comparing and potentially leverage few shot prompting to enhance this part.
> > >
> > > My rating remains the same.

---

### Official Review · Reviewer_FkLi · 2023-10-30

**Soundness:** 3 good
**Presentation:** 3 good
**Contribution:** 3 good
**Rating:** 6
**Confidence:** 3

**Summary:**

The paper introduces a neurosymbolic framework 'LILO' for learning interpretable libraries of reusable code abstractions.
* A dual-system synthesizer that uses both LLM-guided search and enumerative search.
* A compression module based on 'STITCH' that extracts common abstractions from synthesized programs.
* An auto-documentation module that adds human-readable names/descriptions to make abstractions more interpretable.

**Strengths:**

**Originality**:
* The integration of LLMs and symbolic program compression is a novel approach toward library learning.
* Propose auto-documentation for improving abstraction interpretability.

**Quality**:
* Well described architecture.
* The analysis and experiments are comprehensive.
* Comparision to multiple baselines on 3 program synthesis benchmarks demonstrate the advantages of the proposed framework.

**Clarity**:
* The overall framing and individual components of LILO are well motivated.

**Significance**:
* The proposed 'LILO' provides a generalizeable blueprint for integrating language models with formal program synthesis.

**Weaknesses:**

* In Table 1, the proposed 'LILO' framework has substantially higher standard deviation than the baseline across three domains. It indicates LILO's performance varies more widely across different runs, suggesting less stability in this field.
* The dual-system search module combines LLM-guided and enumerative search, yet the tradeoff and relative contributions are not deeply analyzed.
* The proposed 'LILO' is strongly based on LLM and it mentions using different models like Codex, gpt-3.5-turbo, gpt-4, but did not provide direct comparison of the their performance.

**Questions:**

* Please address my concern in the above weakness section.
* In Figure 10 caption D, the authors mentions LLM output are sampled without clearly specify the filtering/selection standard.  I advise the author can give a more detailed explanation.

---

> ### Author Response · Authors · 2023-11-16
> **Thank you for your review!**
>
> We are glad that you found this work to be “novel,” “well-described” and “well-motivated;” and that you felt our experiments and analysis were “comprehensive.” We are also delighted that you agree LILO provides a “generalizable blueprint for integrating language models with formal program synthesis,” as this is exactly what we set out to accomplish with this work.
>
> Thank you for your review and comments! Please allow us to address some of the points that you raised below:
>
> **In Table 1, the proposed 'LILO' framework has substantially higher standard deviation than the baseline across three domains. It indicates LILO's performance varies more widely across different runs, suggesting less stability in this field.**
>
> Part of the novelty of our method is in the integration with a LLM, which inherently introduces variability across runs. In response to your comments and those of the other reviewers, we have updated Table 1 to more transparently present this variability by underlining all results that fall within $1\sigma$ of the best (bolded) result.
>
> While some of the online synthesis conditions approach LILO’s performance, we note that none of the results from offline synthesis (Table 1, bottom half) fall within $1\sigma$ of the best (bolded) result. Thus, **we are confident that our results indicate that LILO meaningfully outperforms the other models considered in offline synthesis.**
>
> We also would like to highlight that $\sigma$ is computed with respect to relatively small $N=3$ runs due to the computational cost of each full run of online program synthesis, so it is not a very tight approximation of the variability in performance. Prior work (e.g., DreamCoder, LAPS) does not  report this variability as part of the main results, but we felt it important to include some kind of variability estimate in our Table 1.
>
> **The dual-system search module combines LLM-guided and enumerative search, yet the tradeoff and relative contributions are not deeply analyzed.**
>
> Thank you for the suggestion. We have added two new baselines to the “offline synthesis” section for the LLM Solver and LLM Solver (+ Search). While these are primarily intended to address a specific concern from Reviewer oPbs, these baselines also help to further pinpoint the relative contributions of LLM-guided search, enumerative search, and compression within LILO.
>
> Beyond these new baselines, our existing results set already gives significant attention to the question of the relative contributions of the two search methods. As a recap, we include the following baselines/ablations as part of our original experiments:
>
> - LILO (No Search)
> - LLM Solver (+ Search)
>
> This pair of conditions (removing enumerative search from LILO / adding enumerative search to the LLM-only baseline) is directly targeted at analyzing the tradeoffs and relative contributions of the two search methods. We discuss these findings on p. 7 (”To isolate the effects of search…”). In addition to the quantitative results, we also mention several qualitative examples of tasks that are solved with enumerative search but not LLM-guided search (e.g., how to draw a “snowflake” or “staircase” in the LOGO domain; see Fig. 5).

---

> ### Author Response · Authors · 2023-11-16
>
> **The proposed 'LILO' is strongly based on LLM and it mentions using different models like Codex, gpt-3.5-turbo, gpt-4, but did not provide direct comparison of the their performance.**
>
> While we would have liked to be in a position to systematically benchmark various LLMs for LLM-guided synthesis, as we discuss in Footnote 1, cost considerations required us to restrict our analysis to Codex.
>
> As a concrete example, based on our numbers in **Appendix C.1, Table 6**:
>
> - An equivalent benchmark of **gpt-3.5-turbo** would have cost **$1,272.36** at the current price of $0.002 / token.
> - An equivalent benchmark of **gpt-4** would have cost **$38,170.80** at the current price of $0.06 / token.
>
> In contrast, we accessed Codex through OpenAI’s free beta program for researchers, which saved thousands of USD over the project lifetime and afforded higher rate limits than paid GPT models.
>
> As discussed in implementation details, we were able to make use of gpt-3.5-turbo and gpt-4 for AutoDoc, which requires orders of magnitude fewer queries. This is why these models are mentioned in the paper, but we make no claims about having systematically benchmarked these other models for synthesis. We also do not think that exhaustive and costly benchmarking of an ever-changing set of foundation models should be a standard for publication at a conference such as ICLR.
>
> **In Figure 10 caption D, the authors mentions LLM output are sampled without clearly specify the filtering/selection standard. I advise the author can give a more detailed explanation.**
>
> Thank you for the attention to detail. This filtering/selection pipeline is already described in Section 3 ("For each completion, we run parsing, type inference, and execution checks to identify valid programs that solve the target task.”). To provide more detail, our post-processing pipeline for LLM completions contains the following steps:
>
> Given a `completion: str`
>
> 1. Check whether `completion` parses to a valid program in the DSL. If not, discard.
> 2. Check whether we can perform type inference on `completion`. If not, discard. At this point, `completion` represents a valid, well-typed `program` in the DSL.
> 3. Check whether `program` satisfies the target task specification (i.e., execute it against all I/O examples). If it does, add `program` to the set of task solutions.
>
> If there are any specific aspects that you would like more detail on, please let us know. Our code on GitHub (currently anonymized for review) will also provide a clear reference for how this pipeline is implemented.

---

> > ### Comment · Reviewer_FkLi · 2023-11-20
> >
> > Thank you for the detail information. I have increased the score.

---

> > > ### Author Response · Authors · 2023-11-21
> > >
> > > Thank you so much -- we're happy we were able to address your questions!

---

### Official Review · Reviewer_oPbs · 2023-10-31

**Soundness:** 4 excellent
**Presentation:** 4 excellent
**Contribution:** 3 good
**Rating:** 8
**Confidence:** 5

**Summary:**

LILO is a system for library learning that leverages large language models. It iterates through a series of modules that allow it to automatically discover a collection of interpretable and documented abstraction functions that help solve held-out tasks across three domains: string regexes, turtle graphics, and visual-question answering. First, given the current version of a DSL, LILO asks an LLM to infer programs that solve prompted input tasks (this step is augmented by an enumerative search). Then, an abstraction discovery algorithm (STITCH) is employed to identify and refactor common patterns of code use into abstraction functions that are added to the base DSL to form a new library. These abstractions are then presented to an LLM, so that they can be given interpretable names and a doc-string description. Compared with prior library learning techniques (DreamCoder) LILO is able to solve more held out tasks with its discovered abstractions.

**Strengths:**

The paper is well-written and proposes a compelling method for a difficult and important problem. The methodology employed is sound and clearly described, even without a reference implementation much of the system seems reproducible. On the whole, the experimental design is reasonable and comprehensive, modulo a few missing conditions (detailed in the next section). I was pleasantly surprised by the detailed computational efficiency analysis at the end of the appendix, it was a well-formulated and interesting addition to the paper.

In terms of contribution, the paper situates itself as a complete system that offers an improvement over DreamCoder across a varied collection of tasks; I think the evidence the paper presents on this point is compelling. There are two main insights/observations that support this improvement. (1) Given a task description, LLM’s can usefully solve inductive programming tasks and (2) the success of (1) is dependant on operating over a DSL with “good documentation”, e.g. without semantic names, even otherwise helpful functions can contribute to model obfuscation. However, in some ways the most surprising results from the paper are that, even without library learning, LLMs can solve these domain-specific tasks much better than prior works (e.g. for regex, comparing DreamCoder to and LLM Solver that infers with respect to a base DSL offers a large improvement).

The paper notes that invoking an LLM at inference time can actually be more computationally efficient than enumerative search, which is an interesting observation. That said, I think the strongest experimental evidence in favor of the proposed method is in the bottom half of Table 1, which shows that LILO produced libraries outperform other libraries when only enumerative search is used. This is a clever, although not perfect, way of evaluating how well-suited each discovered library is for a particular domain.

**Weaknesses:**

My main concern is that while LILO does offer an improvement over DreamCoder, I’m not convinced the claimed reasons for the improvement are conclusive based on the experimental design.

For instance, the version of LILO without enumerative search actually does worse than only using LLM solver for all three of the listed domains, even though LILO has access to an abstraction discovery method (i.e. STITCH). So then, is the evidence that LILO w/ search outperforms LLM Solver+search related to LILO’s use of LLMs (to infer programs and document abstractions) or its use of STITCH to identify common patterns of code-use. On this note, the comparisons between the LLM solver+search variant and LILO appear a bit more murky than the narrative reads; based on the listed standard deviation numbers it’s unclear if there are statistically significant differences in any of these cases.

To better separate out how the contributions of LILO (LLM program induction in the context of iterative library learning, along with auto-documentation of abstractions) affect system performance, I think the following conditions should be considered:

- (A) Base DSL + STITCH
- (B) LLM Solver+Search+STITCH

Condition (A) would be an ablated version of LILO, where no LLM calls are made. “Wake” would be performed through enumerative search, and STITCH would still be used to generate new libraries. This would be similar to a DreamCoder + STITCH integration, where STITCH replaces DreamCoder’s sleep phase. From my perspective, this ablation condition is critical to supporting the claims of the paper – at present, its possible that the delta improvement between LILO and Dreamcoder is due entirely to LILO’s use of STITCH, and ablating out of the use of LLMs in this manner would provide a definitive answer on this point.


Condition (B) would first employ the LLM Solver+Search for a set amount of iterations, and then would run STITCH over the final set of inferred programs to output a new library. This would be important to support the implicit claim of the method that library learning needs to be performed in the intermediate stages of the algorithm. If LILO outperforms this ablation on offline search performance (e.g. bottom half of Table 1), then it would be a strong mark in favor of the method.

**Questions:**

While I think the paper does a good job of validating that AutoDoc is really helpful in getting the LLM to be able to employ the discovered abstractions, I’m a bit curious to know more about how mistakes in documentation might affect performance (as a few mistakes from AutoDoc were noted). Would it be possible to compare AutoDoc generated documentation versus “oracle” documentation (e.g. expert provided)? Even if done just once, it could provide some useful insights into how close the AutoDoc module is to human performance, and whether the LLM solver that gets “expert” documentation would be any better than the one generated by AutoDoc.

# Minor comments

While I appreciate the goals of the system are to work for 'real-world’ domains, I would consider reframing the introductory statement of the results section, as I wouldn’t consider any of the domains studied in this work to match that description.

I think a bit more effort should be made into clarifying that STITCH is not a contribution of this work – this is clear to a careful reader, but should be made obvious. Additional citations to STITCH in Figure 1 and the respective paragraph in Section 3 would seem appropriate.

It looks like Table 1 has a minor bolding issue in the CLEVR mean column, bottom half. More generally, I’m unsure if bolding only the highest number is the right thing to do in this table, when the standard deviations have a great deal of overlap in many cases.

Maybe I’m misunderstanding this, but the standard inductive program synthesis problem statement at the beginning of Section 2, doesn’t seem to match the problem setting that LILO uses. Instead of being conditioned on a specification with a set of input/output examples that come from the same program, my understanding is that the LLM is conditioned on task-description/program examples (e.g. the in-context examples), and then is prompted with a single new task-description. Is there a reason for this disconnect?

While the performance differences between LAPS and LILO are discussed in A.5, is there a reason that this comparison isn’t put directly into either Table 3 or Table 4?

---

> ### Author Response · Authors · 2023-11-16
> **Thank you for your insightful review!**
>
> We are heartened that you found value in so many facets of this work: that the problem space is “important” and “difficult”; that the methods are “sound”, “compelling”, and “clearly-described”; that the experiment design is “reasonable”; and that the overall paper is “well-written”.
>
> Thank you for your insightful comments and detailed suggestions. **In response to your review, we have:**
>
> - Run two new baselines (See Baseline B, below)
> - Started working on a new AutoDoc evaluation
> - Addressed many of your minor comments, with plans to address all of them in the final submission
>
> # New Baselines
>
> We appreciate your attention to detail around the experiment design and hope to fully address your concerns below. Concretely, the following two experimental conditions were proposed.
>
> ## Baseline A: Base DSL + Stitch
>
> Proposal: This baseline is similar to a DreamCoder + Stitch integration, where Stitch replaces DreamCoder’s compression module. The motivation for this condition is to address the possibility of a confound due to the choice of compression module, which could account for the performance delta between LILO and DreamCoder in the experiments.
>
> *Response:* Your concerns about this confound highlight an oversight in our methods presentation: **Stitch is already used as the compression module for all models reported in the paper. This includes our DreamCoder condition, which corresponds exactly to the proposed “Baseline A.”** Specifically, we swapped out DreamCoder’s compression module to use Stitch while preserving the rest of the architecture. There were two motivations for this change:
>
> (1) To improve the efficiency of the “sleep” phase: for even modestly-sized datasets (i.e., hundreds of programs), Stitch makes the difference between compression running in tens of seconds vs. tens of hours (see the performance benchmarks in [1]; we find similarly striking efficiency improvements on our domains).
>
> (2) To facilitate direct comparability between LILO and DreamCoder and remove the possibility of the confound described here.
>
> **Thank you for highlighting for this oversight—this is precisely the kind of background assumption that would be difficult for the authors to catch in our own work. We promise to clarify this point in our updated submission!**
>
> ## Baseline B: LLM Solver + Search + Stitch
>
> Proposal: This baseline is an extension of the existing LLM Solver (+ Search) condition, where Stitch is run on the final iteration. The motivation for this condition is to produce a library that can be evaluated in offline synthesis so as to isolate the effect of intermediate compression on performance. The reviewer states that if LILO were to outperform this ablation in offline synthesis, it would be a “strong mark in favor of the method.”
>
> ****Response:**** We immediately saw the value of this proposal and ran this baseline over the weekend. For completeness/consistency with the online synthesis conditions, we also ran the analogous version of this baseline for the LLM Solver (no Search) condition, which we expect to be weaker than LLM Solver (+ Search).
>
> After running the experiment, we can now confidently report that LILO outperforms both of these baselines by a good margin ($>1\sigma$) on all domains. As intended, the only implementational difference between LILO and the new LLM Solver (+ Search) baseline with post-hoc Stitch compression is the intermediate compression + documentation steps. Therefore, we agree that this result makes a strong argument in favor of performing library learning in-the-loop.
>
> |  |  | REGEX |  |  | CLEVR |  |  | LOGO |  |
> | --- | --- | --- | --- | --- | --- | --- | --- | --- | --- |
> |  | max | mean | std | max | mean | std | max | mean | std |
> | LLM Solver | 48.60 | 43.00 | 5.17 | 91.26 | 89.64 | 2.02 | 36.04 | 27.33 | 7.56 |
> | LLM Solver (+ Search) | 63.40 | 55.67 | 7.51 | 91.26 | 89.00 | 3.92 | 28.83 | 27.63 | 1.04 |
>
> **We have updated our submission with these new results (Table 1 and Figure 4)—please see the updated PDF!** Thank you for this insightful suggestion—we certainly feel these new baselines strengthen the empirical claims of the paper and hope you feel similarly.
>
> # Additional experiments
>
> **Evaluating the quality of AutoDoc documentation against human experts.**
>
> We appreciated your suggestion to compare AutoDoc outputs against expert-generated documentation. Internally, we had discussed something similar as a follow-up to this paper and were inspired by your suggestion to start working on a limited version of this experiment. So far, we have manually corrected AutoDoc errors (e.g., the kind of issues highlighted in Fig. 5) and and are working on a head-to-head evaluation of downstream LLM-guided synthesis performance. We are not sure whether we will have something ready in time for the review response deadline, but will keep you posted.

---

> > ### Author Response · Authors · 2023-11-16
> >
> > # Minor comments
> >
> > **While I appreciate the goals of the system are to work for 'real-world’ domains, I would consider reframing the introductory statement of the results section, as I wouldn’t consider any of the domains studied in this work to match that description.**
> >
> > Fair point! We will update the submission accordingly to remove that language.
> >
> > **I think a bit more effort should be made into clarifying that STITCH is not a contribution of this work – this is clear to a careful reader, but should be made obvious. Additional citations to STITCH in Figure 1 and the respective paragraph in Section 3 would seem appropriate.**
> >
> > Thank you for the suggestion—we will add these additional citations to Stitch in the locations you suggested.
> >
> > **It looks like Table 1 has a minor bolding issue in the CLEVR mean column, bottom half. More generally, I’m unsure if bolding only the highest number is the right thing to do in this table, when the standard deviations have a great deal of overlap in many cases.**
> >
> > Thank you for spotting this bolding issue—this has been addressed. Regarding overlap in results, this is something that other reviewers have also noted. We’ve gone ahead and underlined all results within 1 standard deviation of the best (bolded) result in each category.
> >
> > **Maybe I’m misunderstanding this, but the standard inductive program synthesis problem statement at the beginning of Section 2, doesn’t seem to match the problem setting that LILO uses. Instead of being conditioned on a specification with a set of input/output examples that come from the same program, my understanding is that the LLM is conditioned on task-description/program examples (e.g. the in-context examples), and then is prompted with a single new task-description. Is there a reason for this disconnect?**
> >
> > While the LLM-guided synthesis module doesn’t condition on the input/output examples, LILO  uses the I/O examples in two other ways:
> >
> > - To compute $P(t \mid \pi)$ via program execution.
> > - To approximate $Q (\pi \mid t)$ by training a neural recognition model for enumerative search to predict a task-conditioned PCFG.
> >
> > In principle, we could also make LLM-guided synthesis (i.e., Eq. 3) conditional on the I/O examples. In practice, implementing example-guided synthesis with current LLMs presents a couple of concrete challenges:
> >
> > ****Representational capacity:**** For text-based domains, there are often many I/O examples, and each example may contain a large, structured object (e.g., JSON encoding a CLEVR scene graph). Including the I/O examples in the prompt therefore tends to consume a large portion of the LLM context window. It’s not clear whether this is worth the tradeoff vs. simply including more task instances, though this is an empirical question.
> >
> > **Representational modality:** For image-based domains, it is not yet clear how best to encode the I/O examples. One approach would be to use a so-called “multimodal LLM”, though these models were still in their infancy when we began the project and are only just beginning to reach commercial availability.
> >
> > In sum, we view this as an exciting future direction that represents a natural extension of our framework but shouldn’t affect the soundness/scope of the theoretical presentation. As you noted, we follow a standard problem definition of inductive synthesis in Section 2. Because LILO does use the I/O examples in these other ways, we felt it best not to deviate from this standard presentation, while also showing (e.g., in Eq. 3 and Fig. 10) that our implementation of LLM-guided synthesis doesn’t make use of this information.
> >
> > **While the performance differences between LAPS and LILO are discussed in A.5, is there a reason that this comparison isn’t put directly into either Table 3 or Table 4?**
> >
> > (Assuming you’re referring to Table 1) This was primarily due to space and other practical considerations—the prior work from LAPS (Table 3) contains many different rows/ablations that one might reasonably want to compare between and cannot all fit inside Table 1. We also note that only a subset of the metrics that we report in Table 1 were reported in prior work. For instance, in Table 3, the *max* column is missing from REGEX and CLEVR, and standard deviations were not reported, so integrating that table into Table 1 would leave some gaps. We reasoned the simplest solution presentationally would be to keep all of the results from this work in one table and preserve the LAPS table in its entirety in the Appendix. If there’s extra space in the submission or if you have strong feelings on the issue, we can consider moving it over.
> >
> > **Lastly, we want to thank you again for your time and engagement, which has already resulted in several concrete improvements to this work.**

---

> > > ### Comment · Reviewer_oPbs · 2023-11-20
> > >
> > > I would like to thank the authors for their detailed responses and engagement! After the rebuttal I am much more positive on the paper: the strong performance against the baselines discussed above (along with the clarification that STITCH is used throughout) clearly demonstrates the value of the proposed method.
> > >
> > > I have updated my score accordingly, and at this point recommend acceptance without reservation.
> > >
> > > My only outstanding thought is that it would while footnote 1 is much more helpful in making the distinction between the original DreamCoder method and the DreamCoder ablation considered in this paper, I'm still a bit apprehensive that many readers might miss this reference. To make sure this key piece of information is not lost, it would seem best to make this point in the main text, somewhere in Section 4 (perhaps the discussion).

---

> > > > ### Author Response · Authors · 2023-11-20
> > > > **Recommendation of acceptance**
> > > >
> > > > Dear Reviewer,
> > > >
> > > > We thank you so much for your prompt response. We're happy that we were able to address your feedback and are delighted that you've chosen to update your score in favor of this strong recommendation for acceptance!
> > > >
> > > > Per your suggestion, we've moved Footnote 1 into the main text in Section 4 and updated the submission PDF accordingly. This point of clarification can now be found at the bottom of p. 6 where the different ablations are introduced. We agree that this is a natural location for this information and thank you for your attention to detail.
> > > >
> > > > Best,
> > > > Authors

---

### Author Response · Authors · 2023-11-16
**First round of responses posted!**

We thank all the reviewers for their encouraging and thoughtful feedback! Overall, reviewers found LILO “well-written” (oPbs, dfJc); “refreshing” (dfJc), “novel,” (FkLi), and “conceptually pleasing” (dfJc). The methods and experiments were described as as “comprehensive” (FkLi, oPbs) and “sound” (oPbs), with one reviewer noting that “even without a reference implementation much of the system seems reproducible” (oPbs). Overall, reviewers characterized the work as a “successful combination of LLM and symbolic tools” (NLqL), providing a “compelling method for a difficult and important problem” (oPbs).

We also grateful to have received a variety of constructive comments, with different reviewers focusing on different aspects of the paper. For this reason, **we have posted individualized responses to each reviewer** and will limit the rest of this general response to address a few high-level matters.

# General responses

****New offline synthesis baselines.**** Reviewer oPbs proposed a new condition for the offline synthesis experiments designed to pinpoint the effects of performing **intermediate** compression during the LILO loop vs. performing post-hoc compression on the tasks solved by the LLM Solver / LLM Solver (+ Search) baselines.  The reviewer states that if LILO were to outperform this ablation in offline synthesis, it would be a “strong mark in favor of the method.” After running the experiment, we can now confidently report that **LILO outperforms both of these baselines by $>1\sigma$ on all domains** (see Table 1 and Figure 4 in the updated PDF)**.** We agree that this result we agree that this result strengthens the overall empirical argument in favor of LILO and thank the reviewer for their contribution.

**Standard deviations in Table 1.** In response to comments from Reviewers oPbs and FkLi, we have updated Table 1 to more transparently present this variability by underlining all results that fall within $1\sigma$ of the best (bolded) result. Part of the novelty of our method is in the integration with a LLM, which inherently introduces variability across runs. While some of the online synthesis conditions approach LILO’s performance, we note that **none of the results from offline synthesis (Table 1, bottom half) fall within $1\sigma$ of the best (bolded) result**. These include the two new baselines added during the response period. We also note that LILO’s $L_f$ consistently finds more solutions *faster* than the other libraries evaluated in offline synthesis. Thus, we are confident that our results indicate that LILO meaningfully outperforms these baselines at library learning.

**Clarifying the relationship of this work with Stitch.** Stitch is featured heavily in this work, as it represents the first application of the algorithmic techniques from prior work (Bowers et al., 2023) to online program synthesis. **In all our experimental conditions—including the DreamCoder baseline—Stitch is used as the compression module.** The main benefits of this modification are (1) to improve the efficiency of the “sleep” phase and (2) to remove a possible  confound (noted by Reviewer oPbs) where performance deltas between LILO and DreamCoder could otherwise be explained by the difference in the compression algorithm. We plan to make this important detail more salient in our methods presentation. We also plan to add additional citations to clarify that the algorithmic contributions of Stitch are prior work.

# Submission update timeline

- We have posted a new version of the PDF containing **a limited set of updates focused on the experiments:**
    - **Added two new baselines to the offline synthesis experiments** requested by Reviewer oPbs (updates to Table 1 and Figure 4).
    - **Updated Table 1 to clarify variability** by underlining all results that fall within $1\sigma$ of the best (bolded) result.
- We’re currently working on incorporating miscellaneous changes requested by different reviewers. **We are aiming to post a full revision 2-3 days before the end of the response period** on 11/22 so as to leave enough time for any follow-up comments.

**Thank you again to all the reviewers for your time and engagement with our work!**

---

### Author Response · Authors · 2023-11-20
**Revisions complete!**

We’ve updated the PDF with the full set of revisions to our paper. We thank the reviewers again for the thoughtful comments and feedback. **Please let us know if there is anything else we can address before discussion closes.**

## Summary of changes

Here is a summary of the changes we made during the revision period in response to feedback from the reviewers:

- Added two new offline synthesis baselines described in detail in **[this thread](https://openreview.net/forum?id=TqYbAWKMIe&noteId=wSj76dwvQs)** (oPbs)
- Updated Table 1 to underline values within $1\sigma$ of the best result (oPbs, FkLi)
- Clarified that Stitch is used as the compression module in all models (see new Footnote 1) and inserted additional citations to the Stitch paper (oPbs)
- More clearly delineated the two search procedures in Section 3 and added a reference to the appropriate appendix sections in DreamCoder for implementation details on neurally-guided enumerative search (NLqL)
- Re-emphasized the self-improvement aspect of AutoDoc via wording adjustments throughout the paper (dfJc)
- Reorganized the Appendix materials into three main sections and added a Table of Contents for reference (general)

**We have also added a new set of experiments (Appendix C.2) evaluating the feasibility of performing compression with LLMs.** For these experiments, we evaluated GPT-4 vs. Stitch in a standalone benchmark where the goal is to generate reusable abstractions for synthesis. Our analysis highlights some challenges we encountered in inducing a SOTA LLM to generate valid abstractions that achieve meaningful compressivity in our domains. These results are intended to reinforce the motivation of LILO as a neurosymbolic framework that uses LLMs for synthesis while delegating refactoring to a symbolic module that has been engineered to handle this more computationally demanding task of compression.

## Summary of individual review responses

In addition to the above, we responded individually to topics raised by the reviewers. We refer to these responses in the threads below:

- [Novelty and contributions of LILO](https://openreview.net/forum?id=TqYbAWKMIe&noteId=AayBnAirkj) (dfJc)
- [Applications to practical problems](https://openreview.net/forum?id=TqYbAWKMIe&noteId=ocFo9adThq) (dfJc, NLqL)
- [Motivations for few-shot prompting vs. finetuning](https://openreview.net/forum?id=TqYbAWKMIe&noteId=7sqQ6Cp0Zt) (dfJc)
- [Costs of benchmarking a broader set of GPT models](https://openreview.net/forum?id=TqYbAWKMIe&noteId=ocFo9adThq) (FkLi)

We thank the reviewers for these discussions and will make ourselves available to reply to any last-minute queries in the remaining days before discussion closes.

---

### Meta-Review · Area_Chair_ph18 · 2023-12-05

**Metareview:**

The paper introduces LILO, a neuro-symbolic framework for library learning, integrating large language models (LLMs) with symbolic program compression. LILO iteratively synthesizes code, refactors it using STITCH, and documents it with LLMs, creating reusable libraries. Evaluated on tasks like CLEVR, REGEX, and LOGO, LILO shows advancements over traditional LLMs and DreamCoder. Strengths include its novel integration of LLMs and symbolic tools and improved performance across varied tasks. Weaknesses are its higher variability in performance across runs and limited exploration of the trade-offs between LLM-guided and enumerative search. Another concern is its applicability to real-world problems, as the tasks are somewhat contrived and domain-specific.

**Justification For Why Not Higher Score:**

Despite LILO's strong performance and novel approach, its real-world applicability remains a concern. The tasks, while canonical, are somewhat synthetic and focused on lambda calculus, limiting broader implications.

**Justification For Why Not Lower Score:**

Despite its limitations, the paper presents a significant step forward in neurosymbolic program synthesis. The novel integration of LLMs with symbolic program compression, the successful application of STITCH in an online setting, and the continual self-improvement via auto-documentation are notable advancements. The authors’ responses to reviewers show substantial engagement and improvement of the work, justifying acceptance over rejection. The constructive feedback loop and subsequent revisions have clearly elevated the paper above the threshold for acceptance.

---

### Decision · Program_Chairs · 2024-01-16

Accept (poster)